# Dengue virus co-opts innate type 2 pathways to escape early control of viral replication

Chathuranga L. Fonseka[1,2], Clare S. Hardman [1], Jeongmin. Woo[1,3], Randeep Singh[1], Janina Nahler [1], Jiahe Yang[1], Yi-Ling Chen [1], Achala Kamaladasa[4], Tehani Silva[4,5], Maryam Salimi[1], Nicki Gray[1,3], Tao Dong[1,6], Gathsaurie N. Malavige [4,7] & Graham S. Ogg [1,6,7✉]

Mast cell products and high levels of type 2 cytokines are associated with severe dengue disease. Group 2 innate lymphoid cells (ILC2) are type-2 cytokine-producing cells that are activated by epithelial cytokines and mast cell-derived lipid mediators. Through ex vivo RNAseq analysis, we observed that ILC2 are activated during acute dengue viral infection, and show an impaired type I-IFN signature in severe disease. We observed that circulating ILC2 are permissive for dengue virus infection in vivo and in vitro, particularly when activated through prostaglandin $D_2$ ($PGD_2$). ILC2 underwent productive dengue virus infection, which was inhibited through CRTH2 antagonism. Furthermore, exogenous IFN-β induced expression of type I-IFN responsive anti-viral genes by ILC2. $PGD_2$ downregulated type I-IFN responsive gene and protein expression; and urinary prostaglandin $D_2$ metabolite levels were elevated in severe dengue. Moreover, supernatants from activated ILC2 enhanced monocyte infection in a GM-CSF and mannan-dependent manner. Our results indicate that dengue virus co-opts an innate type 2 environment to escape early type I-IFN control and facilitate viral dissemination. $PGD_2$ downregulates type I-IFN induced anti-viral responses in ILC2. CRTH2 antagonism may be a therapeutic strategy for dengue-associated disease.

[1] MRC Human Immunology Unit, MRC Weatherall Institute of Molecular Medicine, University of Oxford, Oxford, UK. [2] Department of Medicine, Faculty of Medicine, University of Ruhuna, Galle, Sri Lanka. [3] MRC WIMM Centre for Computational Biology, Medical Research Council (MRC) Weatherall Institute of Molecular Medicine, University of Oxford, Oxford, UK. [4] Allergy Immunology and Cell Biology Unit, Department of Immunology and Molecular Medicine, University of Sri Jayewardenepura, Nugegoda, Sri Lanka. [5] General Sir John Kotelawala Defence University, Rathmalana, Sri Lanka. [6] Chinese Academy of Medical Science (CAMS) Oxford Institute (COI), University of Oxford, Oxford, UK. [7] These authors contributed equally: Gathsaurie N. Malavige, Graham S. Ogg. ✉email: Graham.ogg@ndm.ox.ac.uk

Dengue viral infection is one of the most rapidly emerging mosquito-borne viral infections in the world, resulting in a substantial health economic burden in tropical countries. Manifestations of dengue infection can range from asymptomatic infection, dengue fever (DF) to the more severe forms, dengue haemorrhagic fever (DHF) and dengue shock syndrome (DSS). Severe dengue infection is characterised by vascular plasma leakage, leading to depletion of the intravascular volume resulting in haemodynamic shock and haemorrhage causing morbidity and mortality[1]. This phase of vascular plasma leakage is known as the 'critical phase' of dengue disease. It is well established that subsequent infection (secondary heterologous infection) by another serotype can increase the risk of developing severe dengue[2–5]. In fact, epidemiological studies have shown that acute cases of severe dengue, including DHF and DSS, occur most frequently in secondarily infected patients[6]. Though the exact mechanism for this is not well understood, it is accepted that the phenomenon of antibody-dependent enhancement (ADE) may cause increased pathogenicity. Interestingly, studies have shown that most dengue infections are either asymptomatic or lead to uncomplicated dengue fever (DF), even among patients secondarily infected with a heterotypic dengue serotype[7]. Symptomatic cases appear in only 1–5% of infections[4] and of these, severe disease is seen in ~1% of dengue cases. However, in symptomatic disease mortality in severe cases can be high, up to 20%[8] depending on the endemicity. An epidemiological study of a large number of infants born to dengue-seropositive mothers showed that, although ADE could be transmitted vertically, it did not correlate with the incidence of DF and DHF. The infants who developed DHF did not show significantly higher frequencies or levels of DENV3 ADE activity compared to symptomatic infants without DHF[9]. Thus, it seems that ADE may not fully explain severe disease in dengue patients. Additionally, even during widespread epidemics of dengue fever, the proportion of cases that progress to DHF is small, ranging from <0.5% to 4% of all cases, indicating that other factors contribute to disease severity[10]. Therefore, it is prudent to investigate other mechanisms which may drive severity in dengue disease.

Innate lymphoid cells (ILC) are principally tissue-resident professional cytokine-producing cells, enriched in barrier and mucosal tissues[11]. ILC lack antigen-specific rearranged receptors and they mirror the T-helper subsets in transcription factor dependence and cytokine production. Human ILC2 have been identified in the blood, skin, gut and lung tissue[12] and constitute ~0.1% of blood lymphocytes[11]. ILC2 are the predominant ILC population within the skin and are among the first cells to respond in cutaneous immune responses. ILC2 are present in healthy skin and are enriched in individuals with allergic inflammation such as in atopic dermatitis[13–15]. It has been shown that ILC2 are activated and have enhanced type 2 cytokine production in murine models of dermatitis[14,16–19]. Functionally, ILC2 are potent producers of type 2 cytokine such as IL-4, IL-13, IL-5, GM-CSF and are defined by the presence of CRTH2 and CD127 (IL-7Rα) receptors[20]. ILC2 are activated by alarmins such as IL-33, IL-25, TSLP[14] and also by lipid mediators released by mast cells such as $PGD_2$ and $LTE_4$[21,22]. $PGD_2$ acts on ILC2 through the CRTH2 receptor and is predominantly released by mast cell degranulation which occurs in allergic disease and in viral infections such as dengue[22,23]. Additionally, ILC2 have been found to be activated in respiratory viral infections caused by Respiratory Syncytial virus (RSV) and Influenza virus. In RSV infection, ILC2 act as a driver of the type 2 response stimulated by epithelial-derived IL-33 leading to mucus production and airway dysfunction[24]. In influenza infection, ILC2 play a role in epithelial regeneration and tissue remodelling through production of amphiregulin[25]. Furthermore, IL-33 induced activation of ILC2 promotes wound healing through improved epithelial repair[26]. Additionally, it is well established that ILC2 play a role in helminth worm expulsion[27].

The balance of type 1 and type 2 immunity in dengue has been suggested as a contributing factor to severe dengue disease[28,29]. It was observed that while a type 1 dominant immune response is associated with DF, a type 2 immune response is associated with the severe form of dengue, DHF[28]. Previously, Th2 cells were widely accepted as the exclusive source of type 2 cytokines. However, studies have since discovered that ILC2, which akin to Th2 cells require the transcription factor GATA-3, produce large quantities of type 2 signature cytokines[30]. Supporting type 2 cytokine predominance in dengue infection, it has been shown that ex vivo stimulated human ILC2 secreted significantly higher IL-4 and IFN-γ levels, with a trend of secreting higher IL-13 in DHF than DF patient samples[31]. Another study showed that ILC produce higher IL-4/IL-13 or GATA3+ in dengue infection with warning signs compared to dengue infection without warning signs[32]. Additionally, dengue patients who had lower DENV-NS3 specific IFN-γ production by T cells since the onset of illness, before development of 'critical phase', were significantly likely to subsequently develop DHF[33,34]. Also, lung ILC2 from mice challenged with H1N1 virus subtype, containing a functional PB1-F2 virulence factor, induced a dominant IL-13 + CD8 T cell response, regardless of host IFN-γ expression[35]. Considering that the majority of patients with both DF and DHF have very low magnitude dengue peptide-specific Th1 T cell responses[34], and helper and effector T cells are broadly depleted[36], there could be a state where type 2 responses are dominated by ILC2 with relevance to early control of viral replication.

## Results

**In vivo ILC2 are activated and infected during acute dengue infection.** To investigate a possible role for ILC2 in dengue virus infection we performed RNA sequencing of ILC2 isolated from peripheral blood (Supplementary Fig. 1a) of individuals with DF, DHF and healthy individuals (HC) of Sri Lankan ethnicity.

Firstly, an unsupervised clustering approach was undertaken using principal component analysis to assess clustering of genes based on overall transcriptomic profile with respect to the patient groups. We observed that ILC2 of healthy individuals show a markedly different gene expression profile compared to the patients with DF and DHF (Fig. 1a, b)[37]. We found that HC group was clustered separately from dengue disease groups while there was no notable clustering difference between the two dengue disease groups. Analysis of 281 DEGs, represented as a heatmap (Fig. 1b), confirmed a trend of segregation of HC from dengue disease groups. Moreover, this figure reflected the spectrum of disease with a trend of DF samples more similar to HC than DHF samples. Additionally, we found evidence that ILC2 were activated in dengue disease. The genes encoding the key effector receptors *PTGDR2* (CRTH2) and *CD127* appeared to be down regulated in DF, which is known to occur following ILC2 activation (Fig. 1c). Also, *CD38*, a surface marker that is known to upregulate with cellular activation and through interferon stimulation[38,39], was significantly upregulated in DF and DHF. Moreover, *IL2RA* (CD25) showed a trend of upregulation in dengue disease (Fig. 1c). These findings indicated that ILC2 are activated in vivo during dengue disease. Additionally, we quantified the percentage of ILC2 within the blood of further donors in the defined clinical groups. We observed that ILC2 percentages from lymphocytes showed a trend to be higher in patients with DHF ($P = 0.05$, Fig. 1d) but notably did not show the depletion known to occur in T cell populations during acute dengue infection. Additionally, a previous study by

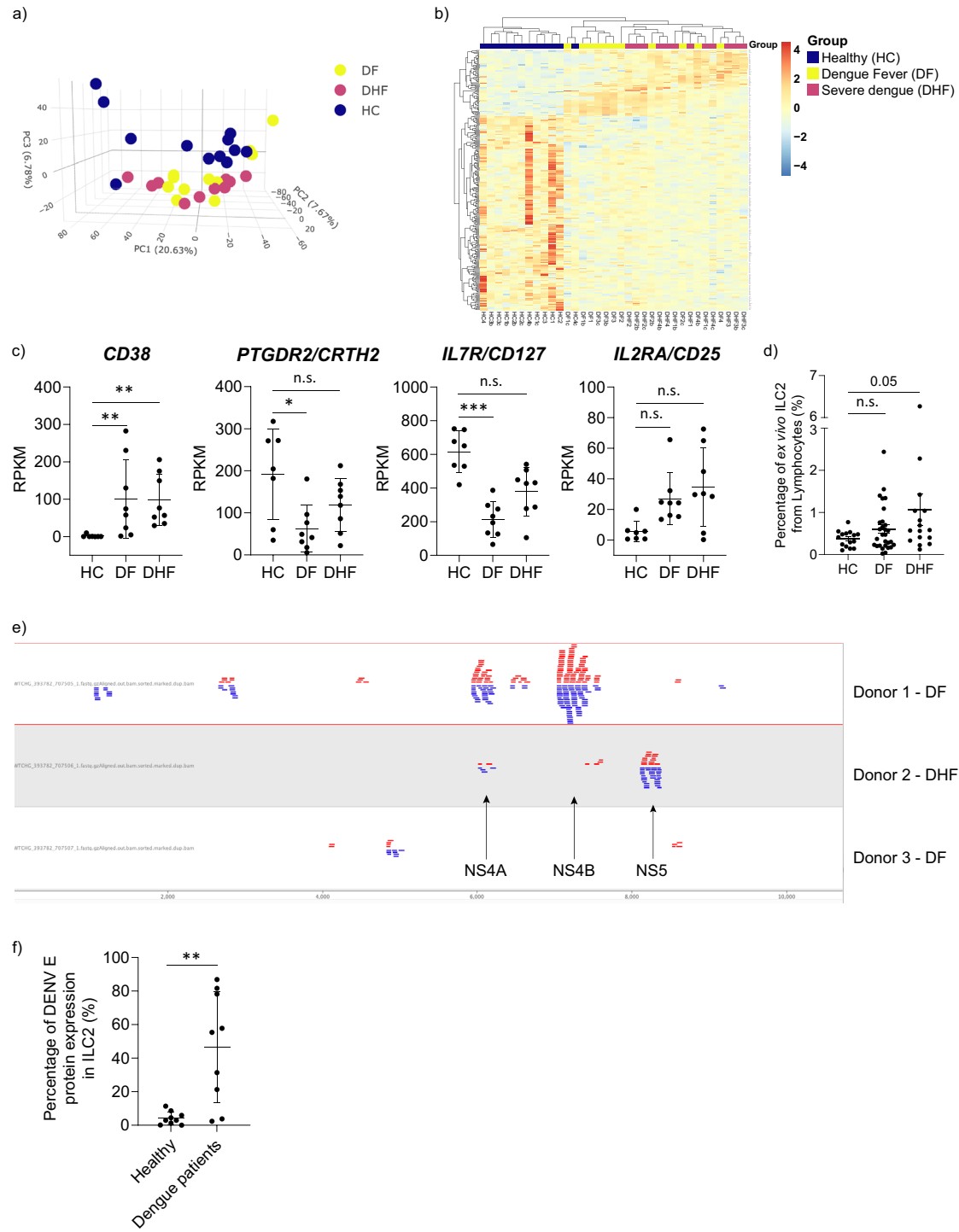

Poonpanichakul et al. showed that there was no significant difference in the frequency of ILC between the febrile phase of dengue infection and convalescence of the same patient[31].

Thereafter, raw RNA sequencing data were aligned against human (hg38) and dengue virus genomes (including all serotypes 1–4) simultaneously to explore whether dengue viral sequences were detectable in ILC2 sequence data. Interestingly, ILC2 from 3 patients (two DF, one DHF) showed detectable dengue sequence of DENV serotype 2. Dengue serotype 2 was the predominant serotype detected in Sri Lanka when the samples were obtained. Most of the reads were originated from the region of 600bp-9000bp of DENV genome, corresponding to NS4A, protein-2K, NS4B, RNA-dependent RNA polymerase NS5 (Fig. 1e). These

data show that dengue virus-derived sequence could be detected in host ILC2.

In addition, ex vivo ILC2 from dengue patients stained positively for intracellular DENV E protein (Supplementary Fig. 1b), with several samples having DENV infection rates as high as 75–85% of the total ILC2 (Fig. 1f). This reaffirmed in vivo dengue viral infection of ILC2.

**In vitro expanded and activated ILC2 are permissive for dengue infection.** Given the apparent activation of ILC2 isolated from dengue patients and dengue virus infection of ILC2 in vivo, next we aimed to determine whether the ILC2 could be directly

**Fig. 1 In vivo ILC2 are activated and infected in dengue infection. a** Principal component analysis and **b** heat map of DEGs (Likelihood ratio test, FDR < 0.05) from ex vivo ILC2 from three separate groups of individuals: healthy individuals (HC, navy blue) ($n = 3$), Dengue Fever (DF, yellow) ($n = 4$) and Dengue Haemorrhagic Fever (DHF, violet) ($n = 4$) in duplicates. DF and DHF groups are categorised using WHO SEARO classification. **c** Activation related gene expression of CD38, PTGDR2 (CRTH2), CD127(IL-7Rα), IL-2RA (CD25) of human blood-derived ILC2 of HC ($n = 3$), DF ($n = 4$) and DHF ($n = 4$), determined by RNA Sequencing and represented in RPKM values. Statistical analysis performed using adjusted $p$ value from the DEG result. $P$ * < 0.05, ** <0.01, *** <0.001, n.s. not significant. **d** Frequency of ILC2 in healthy individuals (HC) ($n = 15$), Dengue Fever (DF)($n = 29$) and Dengue Haemorrhagic Fever (DHF) ($n = 16$). ILC2 were stained as lineage-, CD45+, CD3-, CRTH2+, CD127+ (Statistical significance was tested using one-way ANOVA with Tukey's multiple comparison test. P 0.05, n.s. not significant.). **e** Mapping figure from genome browser depicting matching genes of dengue genome to the raw RNA sequencing data from ILC2 in patients having Dengue fever ($n = 2$) and Dengue Haemorrhagic Fever ($n = 1$). Corresponding proteins for the base pair ranges were depicted. 6376~6756 bp: Non-Structural protein NS4A, 6757–6825 bp: protein 2K, 6826–7569 bp: Non-Structural protein NS4B, 7570~10269 bp: RNA-dependent RNA polymerase NS5. **f** For staining of dengue infection of ILC2, PBMC from HC ($n = 9$) and patients with acute dengue infection ($n = 9$) from Sri Lankan ethnicity were used. Cryopreserved PBMC samples were thawed and stained for ILC2 (Live/CD45+/ lineage/CD3/CD56-/CRTH2+/IL7Rα+) and for intracellular dengue virus envelope (E) protein. Dengue infection was represented as percentage from total ILC2 (Statistical significance was tested using T-test. $P$ * < 0.05, ** <0.01, *** <0.001). All error bars represent mean ± SD.

infected by dengue virus in vitro. Although dengue infection of B cells[40] and monocytes is well documented, there is debate as to whether other cells including T lymphocytes can be infected in vivo. Bulk assays suggest that a proportion of CD2+ cells can be infected although, recent single-cell data suggested that T cells cannot be markedly infected[41,42]. In contrast, recently it has been shown that CD4+ and CD8+ T cells are permissive for dengue infection in ex vivo and in vivo[43]. However, it is unclear whether innate lymphocytes which mirror T helper cell subsets are permissive for dengue infection. Therefore, we aimed to further investigate whether ILC2 are permissive for dengue virus infection.

It is challenging to study mechanisms of ILC2 function ex vivo as they exist in low numbers in peripheral blood. Therefore, to determine whether ILC2 can be infected in vitro, we sorted ILC2 and expanded to sufficient numbers to carry out functional assays. ILC2 from healthy donors were incubated in virus-containing supernatant for 2 h to allow virus adsorption. Subsequently, the unbound virus was removed by washing twice with ILC2 media and the cells were resuspended in ILC2 media containing IL-2. The cells were incubated for 48 h and intracellular dengue envelope protein was detected through flow cytometry. We observed that ILC2 were permissive to DENV2 infection in vitro (Fig. 2a). No intracellular dengue envelope protein staining was detected when cells were exposed to heat-inactivated DENV2. Additionally, dengue anti-envelope protein antibody (4G2) significantly reduced detection of intracellular envelope protein, suggesting infection could be inhibited in the presence of an antibody against dengue virus envelope protein (Fig. 2a). ILC2 infection in vitro was enhanced with increasing multiplicity of infection (MOI) in a dose-dependent manner (Fig. 2b).

Since activation of ILC2 could conceivably occur in type 2 inducing environment, which is considered a risk factor of severe dengue[28], we investigated whether ILC2 activation state could affect the ILC2 infection rate. Alarmins such as IL-33 and lipid mediators such as PGD$_2$ and LTE$_4$ activates ILC2 to produce type 2 cytokines[14,21,22]. We went on to isolate human blood ILC2 from multiple healthy donors, expanded them and activated with IL-33, PGD$_2$ and LTE$_4$ for 24 h, prior to infection with dengue virus and examined intracellular dengue envelope protein using flow cytometry after 48 h. We observed that IL-33, PGD$_2$ and LTE$_4$ stimulation increased DENV E protein staining of ILC2 in a dose-dependent manner signifying dengue infection (Fig. 2c). NS1 protein is the only NS dengue viral protein that is secreted and is efficiently used as a diagnostic marker for early diagnosis of dengue[44]. To demonstrate DENV infection, we measured NS1 protein levels in the supernatant of mock and infected ILC2. Supernatants of DENV infected unstimulated ILC2 and activated

infected ILC2 (with IL-33, PGD$_2$ and LTE$_4$) were 'positive' for NS1 protein while the supernatants from mock ILC2 were 'negative' for NS1 protein (Fig. 2d). PGD$_2$ significantly increased infectivity of ILC2 and acted with IL-33 to increase infectivity of ILC2 compared to dengue virus infection in unstimulated ILC2 (Fig. 2e). ILC2 activation status may therefore influence dengue viral infection, as stimulation via PGD$_2$, LTE$_4$ and IL-33 exerted an increase in DENV infectivity. However, there was no difference of type 2 cytokine levels (IL-13, IL-5 and GM-CSF) between supernatants of mock and infected ILC2 suggesting that the dengue infection may not directly promote secretion of type 2 cytokines from ILC2 (Supplementary Fig. 2).

A number of receptors for DENV have been identified so far suggesting that DENV does not use a unique, specific receptor for its entry into the cell, but recognises and binds to diverse molecules[45]. We stained ILC2 for candidate receptors that could mediate dengue virus attachment and we did not observe any staining for DC-SIGN, MMR, HSP90, HSP70, TIM-3 in activated ILC2 (Supplementary Fig. 3a). DENV infectivity depends on E protein binding to target cell heparan sulphate (HS)[46] which has been characterised as a mediator for DENV attachment and entry[47]. HS is involved with productive dengue infection of endothelial cells[48] and in T cells[43]. From our data, pre-exposure of DENV to heparin, an inhibitor of host HS-virus interactions, led to a significant reduction in infectivity of ILC2, suggesting that HS could be involved in viral attachment and/or entry of DENV to ILC2 (Supplementary Fig. 3b).

Next, we investigated whether ILC2 could support replication-competent dengue virus production. Supernatants from mock and dengue infected ILC2 were collected and Vero cell-based foci-forming assay performed to quantify virus titre in the ILC2 cell supernatant. We found that ILC2 undergo productive infection and release infectious viral particles into the supernatant (Fig. 2f). Interestingly, activation of ILC2, particularly through PGD$_2$, induced secretion of significantly greater infectious viral particles compared to unstimulated ILC2 supernatants (Fig. 2f). This suggests that activated ILC2 contribute to increased viral replication.

For a comparison of infection of type 2 innate cells in dengue infection with the counterparts of the acquired immune system, we performed flow cytometric cell sorting and expansion of Th2 and Tc2 cells over 4–6 weeks. CRTH2 is used as a reliable marker to identify Th2 and Tc2 cells[49,50]. Th2 cells were enriched as CD4 + CRTH2+ cells and the Tc2 cells were enriched as CD8 + CRTH2+ cells (Supplementary Fig. 4). Then, DENV infectivity of ILC2 was compared with pan-T cells (CD3+), Th2 and Tc2 cells. MACS-separated pan-T cells were minimally permissive for dengue infection in vitro (Supplementary Fig. 5a). Comparatively, Th2 and Tc2 cells had significantly higher

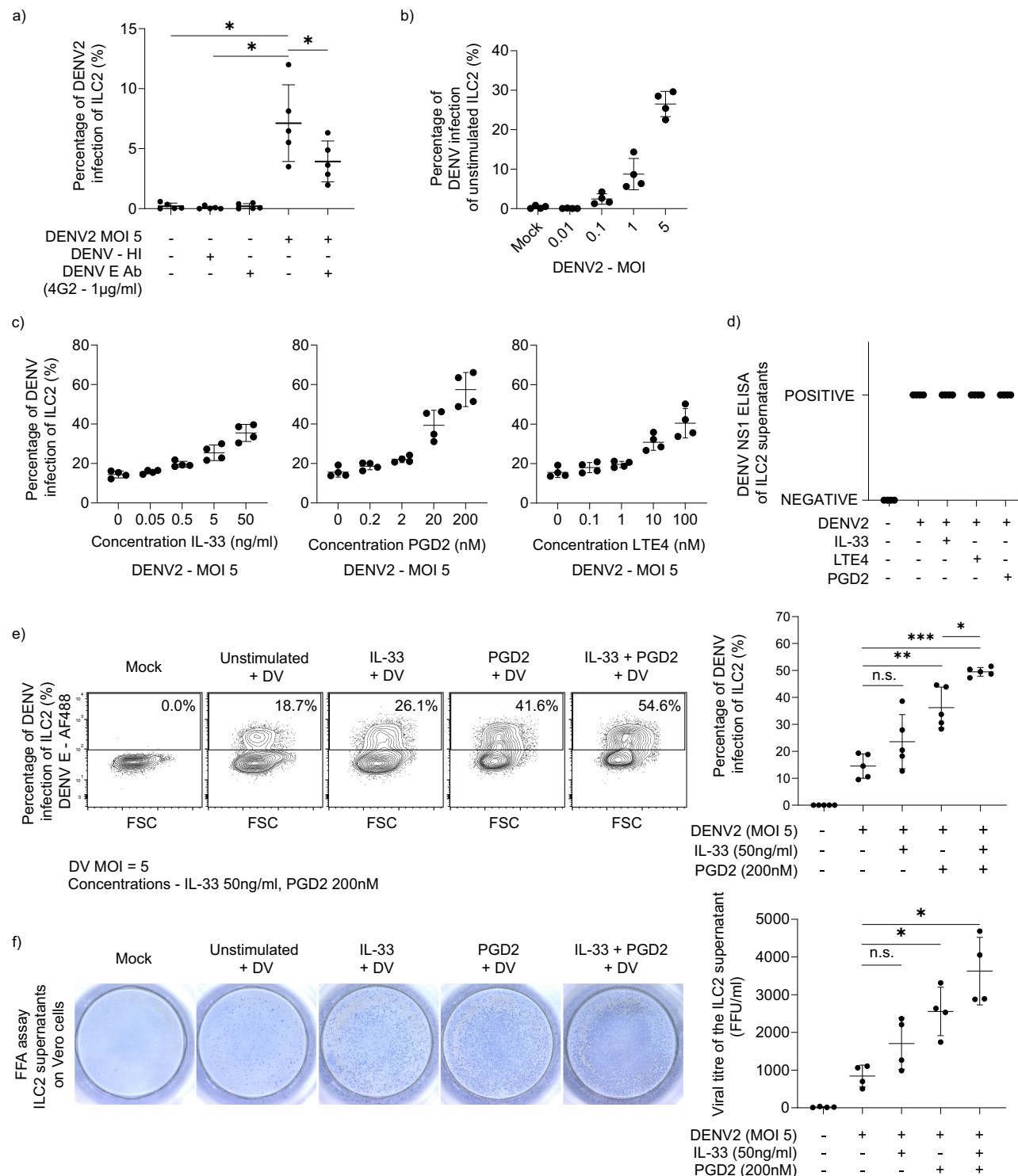

infection rates in an unstimulated state, and the infection significantly increased upon activation with PGD2 (Supplementary Fig. 5b, c) and was significantly reduced with inhibition of CRTH2 receptor (Supplementary Fig. 5b, c). Supernatants from activated infected Th2 and Tc2 contained significantly higher infectious viral particles than unstimulated counterparts (Supplementary Fig. 5d). Compared to Th2 and Tc2 cells, ILC2 had significantly higher levels of dengue infection (Supplementary Fig. 5e) and virus production compared to Th2 or Tc2 or pan-T cells (Supplementary Fig. 5f).

**In vivo ILC2 show impaired type I IFN signature in severe dengue.** The innate immune response is primarily responsible for the control of viral spread during the early stages of infection[51]. The type I IFN response represents a principal effector mechanism of innate immunity and contributes to the control of dengue viral replication and to rapid development of adaptive immune responses that can contribute to elimination of the virus[52,53]. Gene Ontology (GO) enrichment analysis was performed to understand the biological significance of 217 DEGs upregulated in DF and 162 DEGs upregulated in DHF compared

**Fig. 2 Activated ILC2 are more permissive to infection and secrete infectious viral particles. a** ILC2 were exposed to live and heat inactivated DENV2 virus at a MOI of 5 for 2 h. In the blocking condition, dengue envelope protein (clone – 4G2 - 1 µg/ml) was mixed with dengue virus (MOI 5) and incubated for 1 h and was used to infect ILC2. Then, unbound virus was removed by washing cells twice and the cells were plated in fresh ILC2 media containing IL-2 for 48 h. Thereafter, intracellular DENV envelope (E) protein (Clone – 4G2) was checked by flow cytometry. ($n = 4$, Statistical significance was tested using one-way ANOVA with Tukey's multiple comparison test, data representative of 3 independent experiments. $P$ * < 0.05. **b** Unstimulated ILC2 and activated ILC2 (activated with $PGD_2$ for 24 h) were exposed to live DENV2 virus at serial MOI from 0, 0.01, 0.1, 1 and 5 for 2 h. Then, unbound virus was removed by washing cells twice and the cells were plated in fresh ILC2 media containing IL-2 for 48 h. Intracellular DENV E protein was analysed by flow cytometry. ($n = 4$, data representative of two independent experiments). **c** ILC2 were activated through serial concentrations of IL-33 (0–50 ng/ml), $PGD_2$ (0–200 nM) and $LTE_4$ (0–100 nM) for 24 h. Then cells were incubated with dengue virus (MOI 5) for 2 h. Unbound virus was removed and the cells were plated in fresh ILC2 media containing IL-2 for 48 h. Thereafter, intracellular DENV E protein was analysed by flow cytometry. Data representative of three independent experiments. Error bars represent mean ± SD ($n = 4$). **d** DENV NS1 ELISA was performed of the supernatants of mock ILC2, infected unstimulated ILC2 and infected activated ILC2 (with IL-33 (50 ng/ml), $PGD_2$ (200 nM) and $LTE_4$ (100 nM) for 24 h). Data representative of two independent experiments ($n = 5$). **e** Activation of ILC2 with fixed concentrations of $PGD_2$ (200 nM), IL-33 (50 ng/ml), $LTE_4$ (100 nM) was performed for 24 h, followed by infection with dengue virus and then incubated for 48 h. Intracellular DENV E protein was analysed by flow cytometry. ($n = 5$, one-way ANOVA with Tukey's multiple comparison test, data representative of three independent experiments. $P$ * < 0.05, ** <0.01, *** <0.001, n.s. not significant.) **f** Supernatants of infected ILC2 of unstimulated and activated conditions (activated with IL-33 (50 ng/ml) and/or $PGD_2$ (200 nM)) infected ILC2 (with MOI 5) were added on Vero cells and a Foci forming assay (FFA) was performed. Each focus is representative of an infected Vero cell. ($n = 4$, Statistical significance was tested using one-way ANOVA with Tukey's multiple comparison test, data representative of 3 independent experiments. $P$ * < 0.05, n.s. not significant). All error bars represent mean ± SD.

to HC (FC > 1.5 and FDR < 0.05). Our data revealed that DEGs which are upregulated in DF were predominantly associated with the type I interferon response (i.e. GO:006033: type I interferon signalling pathway) (Fig. 3a). Whereas GO analysis of DEGs upregulated in DHF revealed a set of genes that was mainly associated with cell proliferation and DNA replication, and lacked a dominant type I interferon response signature (Fig. 3b). Additionally, individual genes with anti-viral effector functions were analysed in ex vivo sorted ILC2 from HC and individuals with DF and DHF and an upregulation of interferon stimulatory genes (ISGs) such as *ISG15, IFITM3, OAS1, RSAD2, STAT1* was observed in ILC2 derived from individuals with DF but not from those with DHF (Fig. 3c). These genes have been shown to have important functions in dengue fever; for example, ISG15 inhibits viral replication by modifying viral or cellular proteins[54]. OAS1 restricts viral infection by degrading viral RNA in combination with RNase L, resulting in an inhibition of viral replication; and polymorphisms of OAS1 have been shown to associate with dengue patients requiring hospitalisation[55]. The STAT1 pathway acts early and plays a role in controlling initial viral replication[56]. These ILC2 intrinsic findings are consistent with results from two other studies where it was observed that the expression of type I IFN-related genes is clearly upregulated in peripheral blood mononuclear cells (PBMCs) derived from individuals with DF and reduced in patients with DSS[57,58].

To further investigate whether ILC2 are responsive to type I interferons, we treated expanded ILC2 from healthy individuals with exogenous IFN-β (250 U/ml) and observed that anti-viral genes such as *ISG15, IFITM3, OAS1, RSAD2, STAT1* were significantly upregulated in ILC2 (Fig. 3d). Additionally, increasing concentrations of IFN-β had a dose-dependent significant inhibitory effect on dengue virus infection of cultured ILC2 (Fig. 3e). IFN-β protected ILC2 from viral infection and reduced productive infection (Supplementary Fig. 6a, b). It has been shown that plasmacytoid DCs and monocyte-derived dendritic cells (MoDCs) are the main sources of type I interferons in dengue infection[59,60]. While autocrine secreted type I interferons control viral infection in infected MoDCs (Supplementary Fig. 6c), infected ILC2 do not secrete detectable levels of IFN-α (Supplementary Fig. 6d, e) and more likely depend on exogenous type I interferons, and hence risk uncontrolled viral replication. These data suggest that ILC2 are responsive to type I interferon, which limits dengue viral replication; and furthermore, type I IFN signatures are impaired in ILC2 in severe dengue.

**PGD₂ downregulated antiviral responses in ILC2.** Having shown that type I interferons establish an anti-viral state in ILC2 and rescue from infection, we next sought to determine regulatory factors of type I interferon responses in ILC2. $PGD_2$ is a lipid mediator predominantly produced by mast cells, and also produced endogenously by Tc2 and ILC2 following activation[50,61]. Therefore, we investigated whether $PGD_2$ activation modifies the anti-viral state of ILC2.

Stimulation of ILC2 with $PGD_2$ significantly downregulated ILC2 type I interferon receptor, *IFNAR1* expression. *IFNAR1* expression was significantly upregulated with IFN-β treatment (Fig. 4a). Activation via $PGD_2$ also suppressed other anti-viral gene expressions such as *OAS1* (Fig. 4b) and *IFITM3* (Fig. 4c) in a dose-dependent manner. Additionally, $PGD_2$ downregulated IFN-β-induced IFITM3 protein in ILC2 (Fig. 4d). IFITM proteins are known mediators of innate immunity that inhibit viral infection in part by blocking viral entry and have been shown to restrict replication of multiple viruses including dengue[62,63]. Additionally, IFITM proteins are known to restrict ADE in dengue virus infection[64]. With these results, we propose that $PGD_2$ could suppress anti-viral mechanisms in a setting of dengue infection.

Additionally, through use of the CRTH2 antagonist, TM30089, we found that inhibition of CRTH2-mediated activation of ILC2 reduced dengue infection (Fig. 4e). TM30089 significantly reduced dengue virus infection of exogenous $PGD_2$ and IL-33 treated ILC2. IL-33 stimulation induces endogenous production of $PGD_2$ from ILC2[61]. Hence, the effects of endogenously produced $PGD_2$ could be inhibited by pre-treating ILC2 with TM30089. Our data suggest that inhibition of the action of endogenously produced $PGD_2$ from ILC2 reduced dengue virus infection in ILC2 (Fig. 4e). This signifies a functional relevance of endogenously produced $PGD_2$ in ILC2 in the setting of dengue viral infection. Given that lipid mediator $LTE_4$ has been reported to be increased in individuals with severe dengue[65], we investigated whether inhibition of $LTE_4$-mediated ILC2 activation had a similar effect. We observed that montelukast, a leukotriene receptor antagonist which demonstrates selectivity to the cysteinyl leukotriene receptor type-1, can inhibit exogenous $LTE_4$-mediated dengue virus infection of ILC2. In contrast to endogenous production of $PGD_2$, ILC2 did not produce $LTE_4$ endogenously to promote ILC2 activation and dengue viral infection (Fig. 4f). Inhibition of the autocrine action of type 2 cytokines IL-4 and IL-13 through blockade of shared receptor

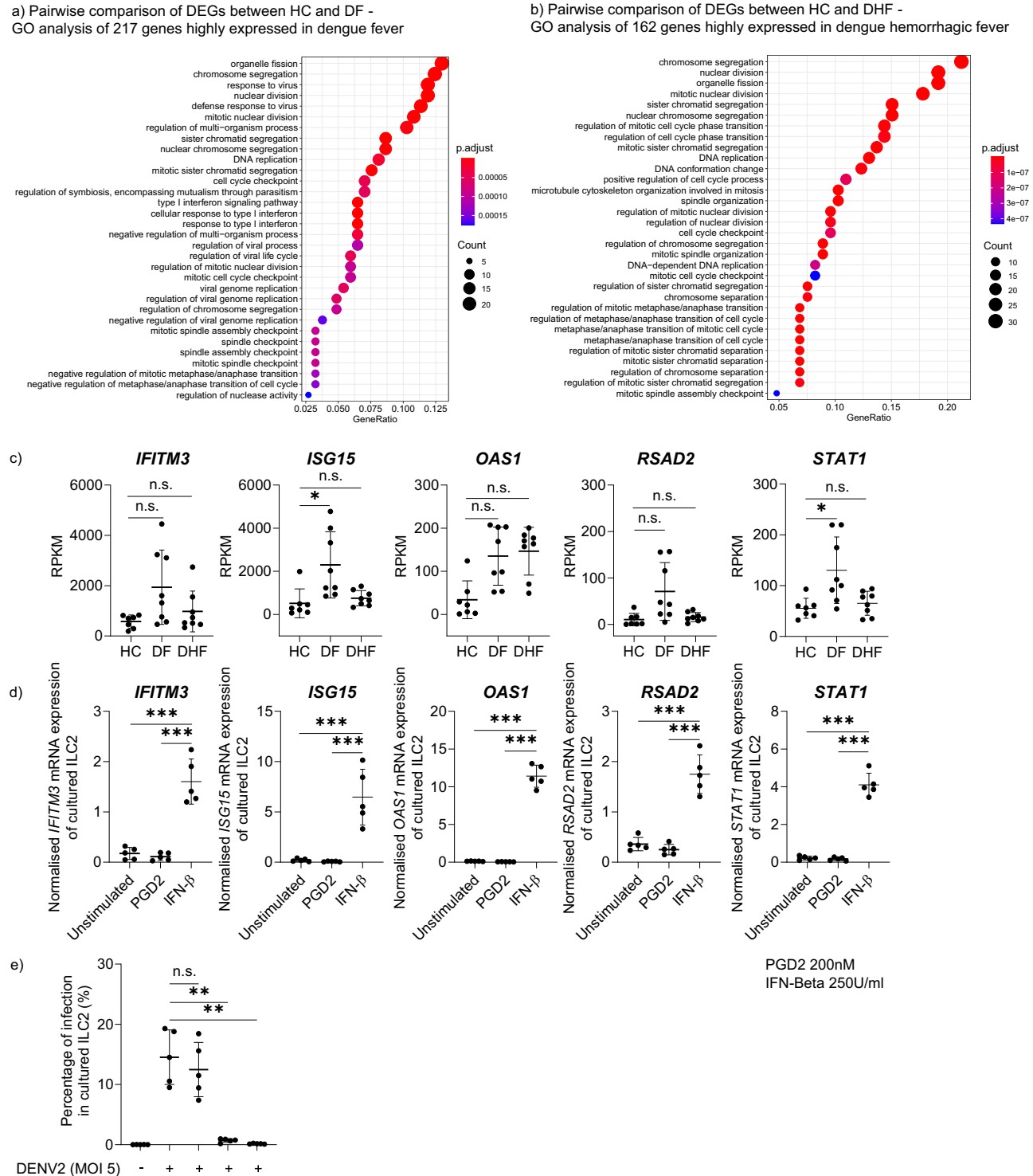

IL4Rα, did not affect the level of dengue viral infection in ILC2 (Supplementary Fig. 7). These data further suggest that inhibition of exogenous PGD$_2$ or LTE$_4$ has a significant effect in reducing dengue infection in ILC2 and endogenously produced PGD$_2$ plays a role in dengue infection in activated ILC2.

**Activated ILC2 increase dengue infection of differentiating monocytes.** Having noted that ILC2 can undergo productive infection, we sought to investigate whether ILC2 could also

influence downstream infectivity of other relevant cells such as monocytes. Cytokine profiling from an outbreak of dengue infection in Taiwan demonstrated that levels of interleukin IL-6, IL-4, IL-13 and GM-CSF were significantly higher in DHF patients compared to DF patients[66]. Major sources of GM-CSF include hematopoietic cells such as T cells, B cells, macrophages, monocytes and it has pleotropic effects on target cell activation and differentiation[67]. In local environments, it is possible that tissue-associated cells such as innate lymphoid cells may play a contributory role. Furthermore, GM-CSF-treated macrophages

**Fig. 3 In vivo ILC2 show impaired type I IFN signature in severe Dengue. a** Gene ontology (GO) enrichment analysis comparing highly expressed genes of ILC2 of healthy individuals (HC) group with DF group (217 genes). **b** GO enrichment analysis comparing highly expressed genes of ILC2 of HC group with DHF group (162 genes). The top ranked GO terms according to gene count, gene number within DEG list, are visualised as dotplots. 'Gene count' is the number of genes enriched in a GO term. Size of the dot corresponds to the gene count. 'Gene ratio' is the percentage of total DEGs in the given GO term. GO terms with clustered genes having higher significant p values are depicted in 'red' colour. **c** Gene expression of type I interferon related genes (*IFITM3, OAS1, ISG15, RSAD2, STAT1*) of human blood derived ILC2 in HC ($n = 3$), DF ($n = 4$) and DHF ($n = 4$), determined by RNA Sequencing and represented in RPKM values. Statistical analysis performed using adjusted $p$ value from the DEG result. P * < 0.05, n.s. not significant. (**d**) Real-time PCR analysis of *IFITM3, OAS1, ISG15, RSAD2, STAT1* gene expression by ILC2 from expanded ILC2 from healthy individuals, following stimulation with PGD$_2$ (200 nM) and IFN-β (250U/ml) for 2 h. ($n = 4$–6, one-way ANOVA with Tukey's multiple comparison test, data representative of 3 independent experiments). Gene expression normalised to *GAPDH*. P *** < 0.001. **e** Expanded ILC2 from healthy individuals were stimulated with IFN-β (2.5-250U/ml), infected with dengue virus (MOI 5) and then incubated for 48 h. Thereafter, intracellular DENV E protein (Clone – 4G2) was analysed by flow cytometry. Statistical significance was tested using one-way ANOVA with Tukey's multiple comparison test, data representative of 3 independent experiments ($n = 5$). P ** < 0.01, n.s. not significant. All error bars represent mean ± SD.

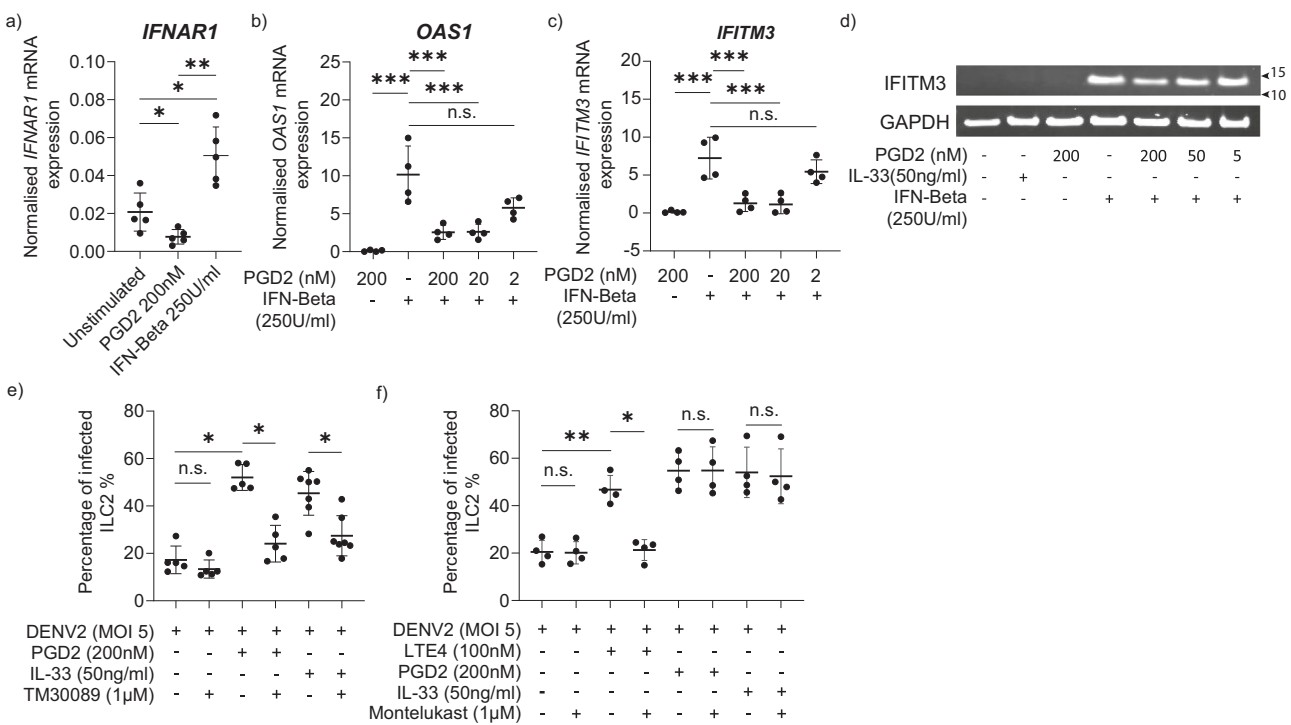

**Fig. 4 ILC2 activation through PGD2 increased the infectivity of ILC2. a** Real-time PCR analysis of *IFNAR1* gene expression by ILC2 following 2 h of stimulation with PGD$_2$ (200 nM) and IFN-β (250 IU/ml). ($n = 5$, one-way ANOVA with Tukey's multiple comparison test, data representative of three independent experiments. P * < 0.05, ** <0.01. Gene expression normalised to *GAPDH*. Real-time PCR analysis of **b** *OAS1* and **c** *IFITM3* gene expression by ILC2 following stimulation with PGD$_2$ (2, 20, 200 nM) for 1 h and then treated with IFN-β (250 IU/ml). ($n = 4$, one-way ANOVA with Tukey's comparison test, data representative of three independent experiments). Gene expression normalised to *GAPDH*. P *** < 0.001. **d** Western blot of IFITM3 protein expression of ILC2 treated with PGD$_2$ (5–200 nM) and subsequently added IFN-β (250 IU/ml). Data representative of two independent experiments (Supplementary Fig. 9). **e** ILC2 were treated with CRTH2 antagonist (TM30089 – 1 μM) for 1 h before treatment with PGD$_2$ (200 nM) or IL-33 (50 ng/ml) for 24 h. ILC2 were infected with DENV2 (MOI 5), and intracellular DENV E protein was assessed. Statistical significance was tested using one-way ANOVA with Tukey's multiple comparison test, data representative of three independent experiments ($n = 5$–7). P * < 0.05, n.s. not significant. **f** ILC2 were treated with Montelukast (1 μM) for 1 h before treatment with PGD$_2$ (200 nM), IL-33 (50 ng/ml), LTE$_4$ (100 nM) for 24 h. ILC2 were infected with DENV2 (MOI 5), and intracellular DENV E protein was assessed. Statistical significance was tested using one-way ANOVA with Tukey's multiple comparison test, data representative of 3 independent experiments ($n = 4$). P * < 0.05, ** <0.01, n.s. not significant. All error bars represent mean ± SD.

produce high levels of IL-1β and pro-inflammatory cytokines on dengue virus infection[68]. ILC2 are potent producers of type 2 cytokines such as IL-13 (Fig. 5a) and GM-CSF (Fig. 5b)[21,22]. It has been shown that type 2 cytokines IL-4 and IL-13 can increase infection of monocytes and monocyte-derived dendritic cells (MoDCs) by upregulating MMR[69,70]. We showed that supernatants from activated ILC2 could indeed enhance MMR expression in differentiating monocytes and MMR expression is dependent on GM-CSF produced by ILC2 (Fig. 5c). Additionally, supernatants from activated ILC2 enhanced dengue infection of monocytes compared to supernatants from unstimulated ILC2

(Fig. 5d). Moreover, we observed that increased infectivity was dependent on ILC2 production of GM-CSF, rather than IL-4/13 (Fig. 5d). Additionally, we observed that mannan, which blocks MMR, DC-SIGN and other receptors with specificity for mannose[69], can inhibit dengue infection in monocytes (Fig. 5e).

**PGD$_2$ metabolite profile throughout the course of illness in acute dengue.** We observed that PGD$_2$ is important in exacerbating ILC2 infection and dengue viral replication. Therefore, we measured concentrations of PGD$_2$ metabolites in the urine of

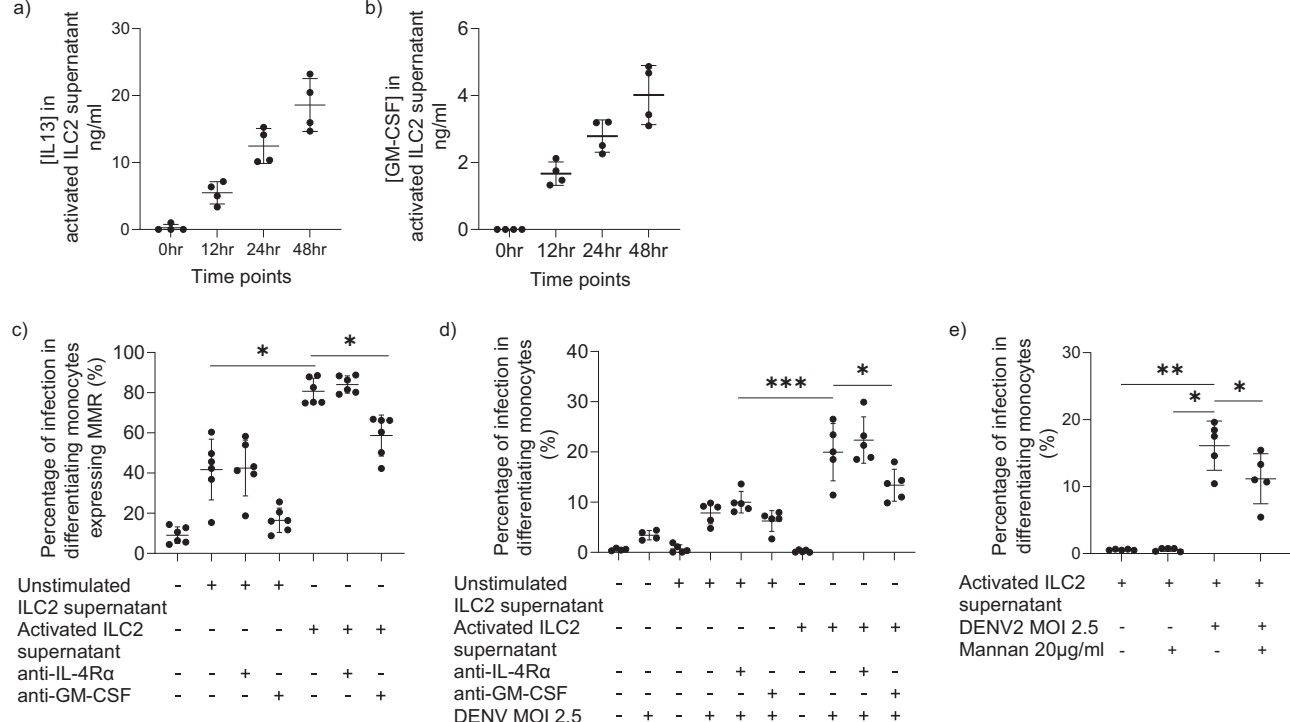

**Fig. 5 Activated ILC2 enhance dengue viral infection in monocytes.** Ex vivo sorted and expanded ILC2 from healthy individuals were stimulated with IL-33 (50 ng/ml), IL-25 (50 ng/ml), IL-2 (100 IU/ml) for 48 h and **a** IL-13 and **b** GM-CSF ELISA assays were performed on the supernatants of ILC2 at 0, 12, 24, 48 hr time points ($n = 4$). To demonstrate the effect of ILC2 supernatant on monocytes, unstimulated ILC2 were incubated with IL-2 (100 IU/ml) and activated ILC2 were incubated with IL-33 (50 ng/ml), IL-25 (50 ng/ml), IL-2 (100 IU/ml) for 48 h and the supernatants were collected. Unstimulated and activated ILC2 supernatants were added to CD14 MACS separated monocytes and **c** MMR expression was measured after 48 h by flow cytometry. In the blocking conditions, anti-GM-CSF and anti-IL-4Rα were added for 1 h before adding supernatants of ILC2. Statistical significance was tested using one-way ANOVA with Tukey's multiple comparison test, data representative of three independent experiments ($n = 6$). $P$ * < 0.05. **d** Unstimulated and activated ILC2 supernatants were added on monocytes and cells were infected with DENV2 at an MOI of 2.5 and intracellular dengue E protein was stained after 48 h by flow cytometry. Statistical significance was tested using one-way ANOVA with Tukey's multiple comparison test, data representative of three independent experiments ($n = 5$). $P$ * < 0.05, *** <0.001. **e** Monocytes were treated with activated ILC2 supernatants for 48 h and infected with DENV2 (MOI 2.5). In the blocking conditions, monocytes were incubated with Mannan 20 μg/ml (Sigma, 9036-88-8) for 1 h prior to infection. Infection rate was measured after 48 h by flow cytometry. Statistical significance was tested using one-way ANOVA with Tukey's multiple comparison test, data representative of three independent experiments ($n = 5$). $P$ * < 0.05, ** <0.01. All error bars represent mean ± SD.

patients with DF and DHF over the course of illness. Inflammatory lipid mediators such as platelet-activating factor and secretory phospholipase $A_2$ levels relevant to the arachidonic acid pathway are known to be higher in patients with DHF, and also urinary leukotriene levels are found to be higher in patients with DHF[65,71,72]. Whilst evidence suggests $PGD_2$ production occurs in platelets, macrophages, T helper cells and dendritic cells, levels are 100–1000 times lower than that synthesised by activated MCs indicating that $PGD_2$ production largely reflects MC activity[73]. $PGD_2$ is an unstable compound and is rapidly degraded and excreted as more stable urinary metabolites[74]. In order to determine the changes in urinary prostaglandin metabolites throughout the course of dengue illness, we measured serial $11\beta$-$PGF_{2\alpha}$ concentrations in the urine of 12 patients with DF and 10 patients with DHF. In patients with DF, urinary $PGD_2$ metabolite levels of $11\beta$-$PGF_{2\alpha}$ gradually declined from day 3 of illness onwards. While in patients with DHF the levels gradually increased to day 6 (Fig. 6a). Additionally, we showed that $11\beta$-$PGF_{2\alpha}$ is significantly higher in the critical phase of patients with DHF compared to the levels 24 h before entering critical phase (Fig. 6b). While $PGD_2$ and its metabolites activate ILC2 function, $PGE_2$ is considered to be inhibitory on ILC2 function[75]; urinary levels of the $PGE_2$ metabolite PGE-M did not show such a trend in patients with DF and DHF (Supplementary Fig. 8). These results in part support the hypothesis that $PGD_2$ plays a role in

DHF progression and the pathway constitutes a promising therapeutic target.

## Discussion

Factors predisposing an individual to severe dengue disease remain incompletely understood. Secondary heterologous dengue infection, obesity, and pregnancy are thought to contribute to morbidity and mortality[76,77], and severe dengue disease has been reported to be associated with atopy, asthma and a type 2 immute state[78,79]. Dengue virus can induce increased endothelial permeability leading to clinically important fluid accumulation in body cavities which is a unique manifestation for a viral illness. A similar phenomenon of increased capillary permeability and tissue fluid accumulation, although with an altered clinical phenotype, is noted in anaphylaxis. ILC2 are shown to be enriched in allergic inflammation in diseases such as asthma and atopic dermatitis[80]. ILC2 are key effectors in the pathogenesis of allergic disorders and here we investigated a previously unrecognised mechanism where ILC2 could contribute to dengue disease pathogenesis.

We showed that ILC2 are activated in dengue fever and permissive to dengue viral infection in vivo and in vitro. Similarly, a recent study showed that ILC1, ILC2 and ILC3 are activated in dengue infection[31]. In addition, our data showed that ILC2

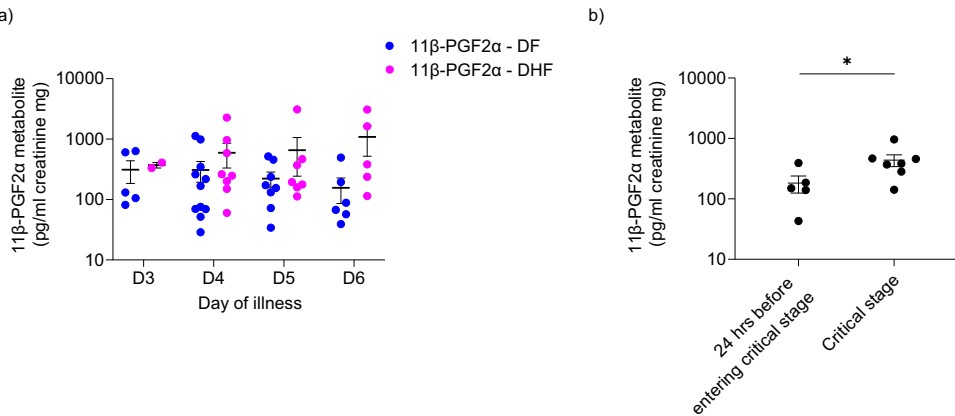

**Fig. 6 Urinary PGD2 metabolite 11β-PGF2α levels in patients with acute dengue.** Urinary **a** 11β-PGF$_{2\alpha}$ levels were measured by quantitative ELISA, in those with DF ($n = 12$) and those who progressed to develop DHF ($n = 10$) in serially collected samples on 3rd, 4th, 5th and 6th days of illness. Error bars represent standard error of mean (SEM) and mean. **b** 11β-PGF$_{2\alpha}$ levels were measured by quantitative ELISA, in those with DHF ($n = 7$) 24 h before entering critical stage and during critical stage. Statistical significance was tested using $T$ test. $P$ * < 0.05. Error bar represent standard error of mean (SEM) and mean.

contributed to highly productive viral replication predominantly when activated by PGD$_2$. We showed that the PGD$_2$ metabolite, 11β-PGF$_{2\alpha}$ is increased in the critical phase of dengue and has a rising trend in severe disease. Mast cells (MC) are the major producers of PGD$_2$[81,82]; it has been shown that MCs are activated and degranulate in dengue[23,61]. Additionally, mast cell products such as leukotrienes and chymases are elevated in dengue infection[65,83]. In dengue models, leukotriene blockade is associated with resolution of vascular leakage[82,84]. Both PGD$_2$ and LTE$_4$ activate ILC2 to produce type 2 cytokine responses[21,22]. Here, we show that these lipid mediators activate innate type 2 pathways in dengue viral infection leading to productive viral replication dissemination directly and indirectly via enhanced monocyte infection.

ILC2 are the predominant ILC2 population in skin and are primed to facilitate type 2 cytokine responses. In contrast to CD4+ cells, the regulatory elements of type 2 cytokine loci of ILC2 are readily accesible in resting ILC2 and are pre-formed even in ILC precursors. In contrast, CD4 + Th cells undergo dramatic remodelling of chromatin after activation[85]. The enrichment of ILC2 during skin inflammation[14] and the capacity of ILC2 to migrate to inflamed skin are important characteristics for a rapid immune response by ILC2[86]. Interestingly, mosquito saliva also affects serum cytokine levels, with the most notable trend being an increase in type 2 cytokines compared to unbitten, control mice relevant to the context of allergic reactions[87]. Therefore, we suggest that early exposure of dengue virus to activated ILC2 at the point of mosquito-bitten skin could have an impact on viral replication. In addition, it has been proposed that ILC2 modulate naive T cell activation, favoring Th2 while suppressing Th1 differentiation[88]. It has been shown that ILC2 are early contributors to immune responses in helminth infections[27] and contribute to an IL-4 dependent enhancement of Th2 responses[89] and orchestrate type 2 cytokine-mediated helminth worm expulsion[27,90]. Additionally, data from murine models suggested that pulmonary ILC2 are more potent than CD4+ T cells in their induction of type 2 cytokines. It is estimated that ILC2 produce ten times more cytokine than T cells on a per cell basis[91,92]. This signifies the capacity of ILC2 to contribute to immune responses despite having a relative scarcity in peripheral blood.

In addition to ILC2, we showed that Th2 and Tc2 cells are permissive to dengue infection in vitro, and PGD2 significantly increased infectivity. Compared to Th2 and Tc2, ILC2 display significantly higher productive dengue infection. In addition, Silveira et al. showed that T cells are permissive to infection in vitro at higher MOI levels than we used. The authors showed that an MOI of 10, T cells had around 15% of infection and at MOI of 100 they showed very high infection rates[43]. Interestingly, ILC2 had the highest productive viral replication compared to pan-T, Th2 and Tc2. In addition, activated ILC2 promote Th2 differentiation through a Notch-GATA3 signal pathway[93]; this may indicate that ILC2 could make a substantially greater contribution to viral replication in vivo, through indirect amplification of viral replication through Th2 cells. Therefore, we believe that ILC2 may contribute to viral dissemination as a result of activation by PGD$_2$ favouring Th2 and Tc2 cell infection and differentiation.

It is known that ILC2 undergo activation and produce a type 2 response during respiratory syncytial virus infection and possibly have a repair role in influenza infection[94,95]. In COVID-19 infection, a higher level of IL-33 was detected in patients consistent with ILC2 activation in addition to detection of higher ILC2 numbers, and comparatively higher levels of type 2 cytokines[96]. In this study, we extend the impact of ILC2 to the pathogenesis of a non-respiratory virus. Type I interferon directly regulates ILC2 and diminishes type 2 cytokine production, proliferation and increases cell death[97]. Reciprocally, we show that the innate type 2 pathway impacts type I IFN mediated control of dengue viral replication within ILC2. It has been observed that IFN-α levels are higher in patients with DF than DHF, irrespective of the infecting serotype, and higher IFN-α levels are found during primary than secondary dengue infection.[60]. These results suggest that an early strong interferon response correlates with a better clinical outcome[60,98]. With gene ontology analysis we showed that genes related to the type I interferon pathway were highly expressed in ILC2 of individuals with DF, while in severe disease this response was blunted. From our in vitro infection data, we observed IFN-β could rescue the ability of ILC2 to control dengue viral replication. Although, MoDCs produce IFN-alpha following dengue viral infection, dengue virus impairs IFN I signalling in DCs[53,99] compared to other viruses, leading to less efficient priming of dengue-specific adaptive immune responses[100]. Dengue virus, therefore, utilises a number of mechanisms to subvert the type I IFN response[101]. Interstingly, PGD$_2$ downregulated type I interferon-induced responses in ILC2, suggesting that type 2 pathways could be an underappreciated escape mechanism to evade type I interferon-

mediated protection during dengue infection. PGD$_2$ acts through both DP1 or DP2/CRTH2 receptors. Werder et al. showed using RSV-infected airway epithelial cell cultures, that DP1 activation up-regulated IFN-λ production, which in turn increased IFN-stimulated gene expression accelerating viral clearance. Additionally, DP2/CRTH2 blockade or DP1 agonism were associated with increased interferon-λ[102]. Moreover, in a neonatal mouse model of severe viral bronchiolitis, CRTH2/DP2 antagonism decreased viral load, immunopathology, and morbidity[102]. Furthermore, it was shown that CRTH2 deficient mice displayed a reduction in exacerbation of allergic airway inflammation and decline in lung function when challenged with viral ds-DNA[103]. Therefore, CRTH2 mediated activation could be a potential viral escape mechanism to suppress protective type I interferon responses and exacerbate viral replication in ILC2 during dengue virus infection. In addition, a randomised clinical trial using oral CRTH2 blockade, observed reduced infections, including influenza in asthmatics[104]. Furthermore, we propose that CRTH2 antagonism would have a therapeutic benefit to reduce the cascade of viral replication.

Mononuclear phagocytes are important in the control of infection and in the development of protective immune responses against dengue virus. Nevertheless, these are FcR-bearing cells which favour dengue viral replication[59]. Viruses can also sabotage the immune system with the use of myeloid cell activation to serve as carriers for transporting the virus from tissues to secondary lymphoid tissues for dissemination to T cells[105]. Interestingly, it has been reported that the viremia in dengue is the result of the massive replication of virus in mononuclear phagocytes in skin and draining lymph nodes, and newly produced viral particles then travel to the bloodstream where they predominantly infect monocytes[106], as well as other immune cell populations, including B cells[40]. We showed that activated ILC2 increase viral infection of monocytes through a GM-CSF dependent, rather than an IL-4/13 dependent mechanism[69,70,107], and may contribute to systemic viral dissemination.

Here we present data that implicate ILC2 in the immunopathology of dengue virus infection. Analysis of human ILC2 derived from dengue infected patient blood samples showed circulating ILC2 can be infected. We showed that PGD$_2$ suppression of protective type I interferon responses in ILC2 can be rescued by exogenous type I IFN. By co-opting a type 2 pathway, dengue virus escapes early type I IFN control and aids viral dissemination directly through enhanced ILC2 infectivity and indirectly through enhanced monocyte infection. With these insights, we propose that inhibition of the PGD$_2$ pathway may be a therapeutic avenue to modulate viral replication.

## Materials and methods

### RNA sequencing analysis of ILC2 among healthy individuals (HC), dengue fever (DF) and dengue Haemorrhagic fever (DHF) patient groups.
Blood samples on days 2–5 of acute dengue illness were taken from four adult individuals of Sri Lankan origin within each group HC, DF and DHF, (during a season where dengue 2 serotype was present) and PBMC were separated by density gradient centrifugation (National Health Service (NHS) National Research Ethics Service (NRES) research ethics committee 14/SC/0106 and Ethics Review Committee, University of Sri Jayewardenepura (741/13)). Aliquots of PBMC were then cryopreserved in freezing media (90% FBS, 10% DMSO) in liquid nitrogen until used.

Cryopreserved PBMC samples were thawed and stained for ILC2 (Live, Lineage-, CD3-, CD45+, CRTH2+, IL-7Ra+) (Supplementary Fig. 1a). Following staining, samples were washed twice with PBS (endotoxin, RNAse free) and resuspended in PBS and sorted at 100 cells/PCR tube directly into lysis buffer (0.2% Triton X-100 solution containing RNase inhibitor) using BD FACS Aria III flow cytometer in single-cell mode. Samples were snap frozen and stored at −80 ˚C. Lysed samples were processed to form cDNA libraries using Nextera XT and SmartSeq2 protocols. RNA-Sequencing was performed in triplicate on an Illumina Hi-Seq 4000 generating 75 bp reads. Following quality control with the fastQC package (http://www.bioinformatics.babraham.ac.uk/projects/fastqc) and trimming poor-quality reads and adaptor sequences, reads were aligned to the human

genome assembly (NCBI build38 (hg38) UCSC transcripts) using STAR (version 2.7.3a) with two pass default mapping mode[108]. Picard tools[109,110] was used to detect duplicate sequences as an additional quality control step and aligned reads were counted with Samtools[111]. Dengue originated reads were analysed with same procedure using combined genome of human and 4 serotypes of Dengue virus (GenBank accession KM204119 for Dengue Virus 1, KM204118 for Dengue Virus 2, KU050695 for Dengue Virus 3 and KR011349 for Dengue Virus 4, respectively) as a reference for mapping. Raw read counts were normalised into reads per kilobase per million mapped reads (RPKM values) using the featureCounts function[112] from the Subread package[113] with default parameters[37] and RPKM values were used to compare gene expression levels of individual genes between healthy and disease groups. RPKM values are represented in duplicates. Sample quality metric and raw read counts were imported in R for further processing. The DESeq2[114] Bioconductor package was used to estimate library size factors, normalise count and perform differential expression analysis. 10,000 most variable genes were used to perform principal component analysis where variance-stabilised transformation expression was used as input. Next-generation sequencing data have been deposited with GSE206648.

Likelihood ratio test was used to analyse differentially expressed genes (DEG) across three groups by comparing 'full' model with the disease status as a factor, to the 'reduced' model which was the intercept. Benjamini–Hochberg multiple testing correction was used to compute false discovery rate (FDR), and genes were considered significantly differentially expressed at <5% FDR. The Wald test was used to perform differential expression analysis comparing DF vs HC and DHF vs HC. Genes were considered significantly differentially expressed if absolute fold change value was more than 1.5 and FDR value was under 0.05.

### GO enrichment analysis.
Gene ontology (GO) terms associated with biological processes that were significantly over-represented for DEGs were identified using the cluster Profiler R package[115]. GO gene sets were tested for overrepresentation in each set of DEGs by computation of enrichment P values with default parameter of 'enricher' function, which implements hypergeometric test. Hypergeometric P values were adjusted for multiple testing using Benjamini-Hochberg correction. The 30 GO terms with highest gene ratio, gene number within DEG list to gene number within background reference, was visualised as dotplots with cluster-Profiler, enrichPlot and ggplot2[116]. The top-ranked GO terms according to gene count, gene number within DEG list to gene number within background reference, are visualised as dotplots. 'Gene count' is the number of genes enriched in a GO term. 'Gene ratio' is the percentage of total DEGs in the given GO term.

### Frequency of ILC2 among healthy individuals (HC), dengue fever (DF) and dengue haemorrhagic fever (DHF) patient groups.
Patients of Sri Lankan ethnicity were enroled within 2–5 days of symptom onset with positive NS1 antigen and/or anti-DENV IgM antibody. Blood samples were obtained on day 2–5 of acute illness. Disease severity was classified using WHO SEARO classification[117]. PBMCs were isolated from healthy adult donors and patients with dengue infection by density gradient centrifugation (Lymphoprep$^{TM}$ density gradient medium). PBMC from Sri Lankan healthy controls ($n = 15$), DF ($n = 29$) and DHF ($n = 16$) patients were stained for ILC2 (Lineage-, CD3-, CD45+, IL7R+, CRTH2+ and c-Kit+) and was represented as a percentage of lymphocytes.

### ILC2 isolation by sorting and cell culture.
PBMC were isolated from healthy adult donors by density gradient centrifugation (Lymphoprep$^{TM}$ density gradient medium). Then T cells and monocytes were depleted from the PBMCs using magnetic-activated cell sorting (CD3 & CD14 MicroBeads, Miltenyi Biotec) and the enriched sample was used for flow cytometric cell sorting. ILC2 were defined as: lineage (CD3, CD16, CD56, CD8, CD14, CD19, CD11c, CD11b, CD123, and FcεRI)−, CD45+, IL7Rα+ and CRTH2+. The ILC2 were sorted into 96-well plates at 100 cells per well. Next, they were resuspended in a mixed leucocyte reaction (MLR) of irradiated PBMCs taken from three healthy volunteers of diverse ethnic backgrounds with IL-2 (200 IU/ml) and PHA (Phytohemagglutinin 2.5 μg/ml). MLR was prepared from PBMCs of the healthy donors which were irradiated with 3000 rad and 100ul of the MLR was added to a 100 μl of cell suspension containing sorted ILC2s. Sorted cells were cultured in RPMI 1640 (Sigma-Aldrich) supplemented with 10% heat inactivated human serum and 2 mM L-glutamine, 10 ml/liter penicillin-streptomycin (Sigma-Aldrich), HEPES, Non-essential amino acids, sodium pyruvate and 2-Mercaptoethanol (ILC2 Media). Then the sorted cells were rested for 5–7 days in MLR suspension and thereafter, half of the medium was replaced with equal volumes of ILC2 media containing 200 IU/ml IL-2. Over the next 4–6 weeks the cells were expanded and after 4–6 weeks, the growing cells were tested by flow cytometry, to ensure a pure population of lineage−CRTH2 + IL-7Rα + ILC2 was obtained. Cells with purity more than 98% were used to conduct experiments.

### Surface staining of ILC2, Th2, Tc2.
For FACS surface staining, the cells were labelled by the following anti-human antibodies: CD3 (SK7; BD), CD19 (SJ25C1; BD), CD123 (FAB301C; R&D Systems), CD11b (DCIS1/18; BD), CD11c (BU15; Abcam), CD4 (A161A1, BD), CD8 (RPA-T8), FcεRI (AER-37 ([CRA-1]), CD14 (MφP9; BD), CD45 (H130), CD56 (B159), CD45 (HI30, BD), CRTH2 (BM16; Miltenyi Biotec), IL-7Rα

(A019D5), and live/dead violet (Invitrogen). The samples were acquired using FACSDiva on LSRFortessa flow cytometer. FlowJo software (FlowJo_v10.7.1) was used for data analysis.

**Th2 and Tc2 isolation by sorting and culture**. PBMCs were isolated from healthy adult donors by density gradient centrifugation (Lymphoprep™ density gradient medium). Then, T cells were separated from the PBMCs using magnetic-activated cell sorting (CD3 MicroBeads, Miltenyi Biotec) and the CD3+ enriched sample was used for flow cytometric cell sorting. Th2 cells were defined as CD3 + CD4 + CD127 + CRTH2 + and TC2 cells were defined as CD3 + CD8 + CD127 + CRTH2+ (Supplementary Fig. 4a). Th2 and Tc2 were sorted into 96-well plates separately at 50 cells per well. Cells were resuspended in a mixed leucocyte reaction (MLR) of irradiated PBMCs taken from three healthy volunteers of diverse ethnic backgrounds with IL-2 (200 IU/ml) and PHA (2.5 µg/ml). MLR was prepared as the same methods as done for ILC2. Sorted cells were cultured in RPMI 1640 (Sigma-Aldrich) supplemented with 5% Human serum and 2 mM L-glutamine, 10 ml/liter penicillin–streptomycin (Sigma-Aldrich), HEPES Non-essential amino acid, sodium pyruvate and 2-Mercaptoethanol. Then the sorted cells were rested for 5–7 days in MLR suspension. They were expanded in the same method as ILC2 over the next 4–6 weeks and the growing cells were checked by flow cytometry, to ensure a pure population of CD4 + CRTH2+ and CD8 + CRTH2+ were obtained (Supplementary Fig. 4b). Cells with purity more than 98% were used to conduct experiments.

**Dengue virus generation and propagation**. The virus was propagated using the C6/36 mosquito cell line, concentrated and cryopreserved in aliquots at −80 °C until used. C6/36 cells were cultured in T-75 close cap flasks at 28 °C. When the cells grow to reach 90% confluence, subcultures were prepared by vigorous pipetting of the cells and they were passaged to new flasks. L-15 (Leibovitz's L-15) media with 10% Foetal Bovine serum, 2 mM ʟ-glutamine, 10 ml/liter penicillin-streptomycin (Sigma-Aldrich), 1% HEPES (10% L15 media) were used for C6/36 cell culture.

Dengue virus serotype 2 (Strain SL 5-17-04, #NR-49751) was used to generate further stocks of virus and the viral stocks were aliquoted and stored in −80 °C until further use. C6/36 cells (kindly donated by Prof Gavin Screaton and Dr. J Mongkolsapaya) were counted and plated in T75 close capped flasks (5 × 10⁶ cells in 12–15 ml of 2% L15 media per flask) inoculated with the virus with a MOI of 0.001–0.01 in 2% L15 cell culture media containing 2% Foetal Bovine serum, 2 mM ʟ-glutamine, 10 ml/l penicillin-streptomycin (Sigma-Aldrich), 1% HEPES. Flasks were incubated at 28 °C. Then, the virus-rich supernatants were obtained on day 4 and day 6. Virus-rich supernatants were spun to remove the cells/debris and then concentrated using a 100 kDa molecular weight cut-off spin columns (Vivaspin 20, GE Healthcare) (centrifuged at 3200 g for 30 min) and the concentrated virus stocks were aliquoted and stored in −80 °C until used in experiments. The concentrated virus supernatants were used for infection experiments.

**Infection of cells with DENV2**. ILC2 were infected using dengue serotype 2 virus (Strain SL 5-17-04, #NR-49751). Stocks with virus titre of 10⁵–10⁶ FFU/ml were used. ILC2 were inoculated with dengue virus at an MOI of 5 for 2 h in 96 well plates. Concentrated 2% L15 media (concentrated from supernatants from mock T-75 flasks parallel to virus concentration) was used as the negative/mock control. Dengue antibody blocking was performed by incubating virus supernatant and Dengue envelope protein antibody (4G2 monoclonal antibody - NBP2-52709, Novus Biologicals −1 µg/ml) for 30 min and then performing virus adsorption. Next, the cells were washed twice with ILC2 media to remove the excess or unbound dengue virus and incubated in ILC2 media containing IL-2 (200 U/ml) for 48 h. Infection rate was detected using intracellular envelope protein staining after permeabilizing and fixing cells.

For FACS surface staining to define ILC2, the cells were labelled with the following anti-human antibodies: CRTH2 (BM16; Miltenyi Biotec), IL-7Rα (A019D5), and live/dead violet (Invitrogen) or PE-Texas-Red (Invitrogen).

Intracellular staining of dengue virus-infected cells was performed by permeabilising cells for 20 min using BD Cytofix/Cytoperm™ Fixation/Permeabilization Kit (554714) and cells were washed with permeabilization wash buffer (PWB) and incubated in intracellular dengue envelope antibody (4G2 NBP2-52709) for 1 h at 37ºC. Cells were washed with PWB twice and incubated in 3% FCS PBS buffer. The samples were acquired using FACSDiva on LSRFortessa flow cytometer. FlowJo_10.7.1 software was used for data analysis.

**Foci forming assay to determine viral titres of ILC2 supernatants**. The concentration of the virus was determined by focus-forming assays on Vero-81 cells (donated by Professor Gavin Screaton and Dr. J Mongkolsapaya) and expressed as FFU/ml. Vero cells were grown in T-75 vented cap flasks at 37 °C. When cells reach a confluence ~90%, the cells were trypsinised, centrifuged, counted and propagated to new flasks. For the Foci forming assay, supernatants from mock and infected ILC2 were added on Vero-81 cell monolayers in 96 flat bottom plates in duplicates and incubated at 37 °C with 5% CO₂ for 2–3 days. Then Carboxymethulcellulose (Sigma #C-5013) 100 ml of overlay was added to each well and incubated at 37 °C for 3 days. After 2–3 days the medium in the wells was discarded and the monolayer was washed five times with PBS. Then the cells were fixed with 4% paraformaldehyde, permeabilized with Triton-X100 and washed again with PBS 3

times. To detect foci, clone-4G2 monoclonal antibody which binds to the fusion loop at the extremity of domain II of envelope protein of flavivirus group (NBP2-52709, Novus Biologicals) was used as the primary antibody and HRP conjugated goat anti-mouse IgG HRP (KPL #074-1806) 1:500 in Ab diluent, as the secondary antibody. The plates were developed using the True-Blue Peroxidase Substrate (KPL #50-78-02). All assays were done in duplicate. The infected foci were counted using EliSpot plate reader. Virus titre (FFU/ml) was calculated by number of foci x 10 ×2 divided by the dilution of the virus.

**qRT-PCR**. mRNA extraction was performed from 10⁵ cells/well ILC2, following stimulation for 2 h using a TurboCapture 96 mRNA kit (Qiagen, 72251) according to manufacturer's instructions. Complementary DNA was prepared from mRNA by reverse transcription PCR (M-MLV, Invitrogen). qPCR performed using Taqman probes and TaqMan gene expression master mix. TaqMan probes used are mentioned below.

GAPDH (glyceraldehyde-3-phosphate dehydrogenase) (Hs02786624_g1)
IFITM3 (Hs03057129_s1)
OAS1 (Hs00242943_m1)
ISG15 (Hs01921425_s1)
RSAD2 (Hs00369813_m1)
STAT1 (Hs01013996_m1)
IFNAR1 (Hs01066116_m1)

**Western blot**. Protein was extracted in 1 million ILC2 in each condition using M-PER (Thermo Scientific - 78501) and protein samples were prepared in reducing buffer and boiled for 5 min, analysed by sodium dodecyl sulphate polyacrylamide gel electrophoresis (SDS/PAGE), and transferred to a polyvinylidene difluoride membrane (Invitrogen). IFITM proteins were detected using mouse anti-human monoclonal IFITM3 antibody 1:1000 (kindly donated by Professor Tao Dong's lab). GAPDH (FF26A/F9 - 649202) was used as the loading control (1:1000).

**Effect of ILC2 supernatant on monocytes**. Monocytes were positively selected from the PBMCs using CD14 magnetic beads (Miltenyi Biotech) using MACS separation columns (Miltenyi Biotech). Then, they were treated with supernatants of ILC2 which were activated for 48 h treated with IL-33 (50 mg/ml) and IL-25 (50 ng/ml). For the blocking conditions IL-4Rα (Human IL-4 R alpha Polyclonal antibody – AB-230-NA - R&D Systems - 10 µg/ml) and GM-CSF neutralising reagent (anti-human GM-CSF monoclonal antibody (23B6 - 3480-5N-500 – Mabtech – 5 µg/ml) were incubated for 1 h before adding ILC2 supernatants. After 48 h monocytes with were stained for expression of MMR (15-2, AF488: 321114). After 48 h of incubation they were adsorbed with dengue virus at a MOI of 2.5 and unbound virus was removed by washing twice with media. Cells were plated in GM-CSF (50 ng/ml) and IL-4 (5 ng/ml) for another 36–48 h and intracellular dengue envelope protein was stained by using AF488/PE conjugated anti-E protein antibody (4G2- NBP2-52709). The samples were acquired using FACSDiva on LSRFortessa flow cytometer. FlowJo_10.7.1 software was used for data analysis.

**ELISA of PGD₂ and PGE₂ metabolites**. The 11beta-prostaglandin F2alpha (11β-PGF₂α) and prostaglandin E metabolite (PGE-M) were measured in urine samples collected from acute dengue patients, which were stored at −80 ºC. These were measured using 11β-PGF₂α quantitative ELISA (Cayman Chemical, USA) and prostaglandin E metabolite quantitative ELISA (Cayman Chemical, USA). Both assays were performed in urine diluted at 1:10 ratio using the respective ELISA buffer. As these metabolites were measured in urine and the renal function affects the levels being excreted, the metabolite levels were normalised using urinary creatinine level of each sample. The urinary creatinine levels were measured using the urinary creatinine colorimetric assay (Cayman Chemical, USA) in urine diluted at 1:10 ratio. All assays were performed according to the manufacturer's instructions.

**DENV NS1 ELISA of ILC2 supernatants**. NS1 ELISA (Novateinbio, NR-R10004) was performed on ILC2 supernatants from mock infected and infected ILC2. Unstimulated ILC2 and activated ILC2, activated for 24 h with IL-33 (50 ng/ml), PGD2 (200 nM), LTE4 (100 nM), were infected with DENV2 and incubated for 2 h. Thereafter, the cells were twice washed with media and cell were plated in 96 well plates for 48 h in ILC2 media supplemented with IL-2. Supernatants were collected after 48 h of incubation. DENV NS1 ELISA was performed according to manufacturer's instruction. Supernatants were diluted 1:20 for the experiment. Quality control was performed with positive and negative control samples. Immune status ratio (ISR) was calculated for each sample condition from the ratio of the mean optical density (OD) obtained from the test sample divided by the mean cut-off value. If ISR value >1 was considered as 'positive' for the presence of DENV NS1 antigen. ISR value <1 was considered as 'negative' for the presence of DENV NS1 antigen.

**Statistics and reproducibility**. The one-way ANOVA tests with Tukey's multiple comparison test, T-tests were performed using GraphPad Prism version 6.00

(GraphPad Software). Error bars represent standard deviation or standard error of means as indicated.

Experiments were performed in duplicates and repeated three times to validate reproducibility. Representative figures of 3–6 donors were presented in the manuscript.

**Reporting summary**. Further information on research design is available in the Nature Research Reporting Summary linked to this article.

## Data and materials availability

All raw and processed next-generation sequencing data have been deposited with GSE206648. Source data for main figures are provided as Supplementary Data 1. Reagents used for stimulation and blocking experiments are provided in Supplementary Table 1.

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

## Acknowledgements

We are grateful for funding from the UK Medical Research Council, Commonwealth Scholarship Commission in the UK, NIHR Oxford Biomedical Research Centre, Chinese Academy of Medical Sciences (CAMS) Innovation Fund for Medical Science (CIFMS) and NIH. Also, we thank the staff of the flow cytometry facility of the MRC Weatherall Institute of Molecular Medicine, especially Craig Waugh.

## Author contributions

Conceptualisation: C.F., C.H., G.M., G.O., Methodology: C.F., C.H., J.Y., T.S., J.W., M.S., J.N., R.S., N.G., Y.L.C., A.K., Investigation: C.F., C.H., J.Y., T.S., J.W., M.S., J.N., R.S., N.G., Y.L.C., A.K., Funding acquisition: T.D., G.M., G.O., Project administration: C.H., T.D., G.M., G.O., Supervision: C.H., T.D., G.M., G.O., Writing – original draft: C.F., C.H., G.O., Writing – review & editing: C.F., C.H., J.Y., T.S., J.W., M.S., Y.L.C., R.S., N.G., T.D., G.M., G.O.

## Competing interests

The authors declare no competing interests.
