## [Peer Review File · Communications Biology]

Reviewers' comments:

Reviewer #1 (Remarks to the Author):

The manuscript "Dengue virus co-opts innate type 2 pathways to escape early control of viral replication" by Fonseka et al., describes that ILC2 are activated during DENV infection, especially in individuals developing SD. They show that ILC2s are activated by PGD2, which also increases the susceptibility of ILC2 directly to DENV infection. PGD2 was found to antagonize the anti-viral effects of type I IFN on ILC2. They also show that ILC2 sups increases monocyte susceptibility to DENV infection via GM-CSF. Authors conclude ILC2 are directly implicated in the immunopathology of dengue, either by increasing monocyte susceptibility to DENV infection or by spreading the virus themselves since are susceptible to infection. They propose that inhibiting PGD2 pathway may be a therapeutic avenue to treat dengue.

The manuscript is technically sound, and all experiments seems to have been performed carefully. A significant novelty of this manuscript is that ILC2 is permissive to DENV infection. It has been demonstrated earlier that lymphocytes (T cells) are also permissive to DENV infection (Silveira et al. J Virol 2018 – not cited). The paper by Silveira et al. is important for their discussion about the role of ILCs to be systemic spreaders of DENV, which seems an immature speculation with the data presented so far. Also, implicating ILC2s conclusively with dengue pathology seems a bit early. At most, an association has been established between ILC2 and SD evolution, which confirms a recently accepted paper that demonstrated that higher frequencies of ILC2 are secreting IL4 and IL13, i.e. they are more activated, in SD cases (Quintino-de Carvalho et al, JID 2021, epub ahead of print – also not cited in the manuscript). Interestingly, Quintino-de-Carvalho found that ILC1, ILC2 and also ILC3 show increased activation levels in SD patients. Type I IFN has also been demonstrated to downmodulate ILC2 (Duerr et al., Nat Immuno 2016 – cited) and that type I IFN counters viral infection is long known.

On the other hand, the manuscript demonstrates that ILC2 might modulate monocytes to be more susceptible to DENV infection and here seems to be the most valuable contribution of the work. How this finding is translated to in vivo, still demands further investigation.

Reviewer #2 (Remarks to the Author):

Dear authors,

The manuscript entitled "Dengue virus co-opts innate type 2 pathways to escape early control of viral replication" was reviewed. The authors showed ILC2 were activated in dengue infection and found DENV RNA in ILCs. They demonstrated that ILC2 can be infected with DENV in vitro which can be enhanced by CRTH2 activation by PGD2 and blocked by CRTH2 antagonist. Further, they showed supernatants from activated ILC2 facilitate monocytes DENV infection. They suggested that type 2 environment esp. PGD2 decrease ILC2 IFN anti-viral response and support viral dissemination associated with severe disease.

The manuscript was well written and the finding is novel.

Major comments;

1. Previous published work on ILC in dengue should be cited and the current results should be discussed with previous work:

- <https://www.frontiersin.org/articles/10.3389/fimmu.2021.599805/full>

- <https://academic.oup.com/jid/advance-article-abstract/doi/10.1093/infdis/jiab312/6298546>

2. Although ILC2 was shown to be permissive of DENV infection in this study, their frequency is quite low. The significance /magnitude of its contribution to the over all viremia should be discussed.

3. Beside monocytes, have the authors investigate their contribution to B cell infection as B cells have recently been shown to be another major source of DENV infection?

4. Fig 1A-B: Please incorporating donor information in the figure 1A-B. (For example, Fig 1A = using color for donor / shape for severity. Fig 1B - another bar labelling donor)

5. Fig 2-4: the authors identified the factors (IL13, PGD2, external IFN β) that affect DENV infectivity on ILC2. How these factors affect the function of ILC2 itself? Does it mean that more infected ILC2 leads to more type 2 cytokine production? How much ILC2 contributes type 2 cytokines to the system overall?

6. Figure 3. It would be interesting to compare DEG with paired convalescence sample from the same patients.

7. Fig 5: the authors showed that blocking GM-CSF decreases DENV infection in monocytes. Did ILC2 produce those factors that lead to more infection in monocytes ? (they also mention this in discussion on page 14 line 355), I wonder if the authors see different expression of gene coding for GM-CSF (CSF2) from ILC2 RNA-seq data in the context of severity?

Minor comments:

- Adding tables of DEGs in supplementary material as well as deposited raw data in GEO can be useful for other researchers in the field.

- Legend of Fig 2C: include sample size and error bar calculation

- Graphical abstract showing the roles of PGD2/LTE4/IFN on ILC2 can be helpful

- Fig 1: It would be nice if the author can validate that ILC2 were infected ex-vivo by FACS or RT-PCR.

Reviewer #3 (Remarks to the Author):

This work is to investigate the role of ILC2 in dengue virus infection. However, the data do not support authors claim. The major concerns are as followings:

1) Authors claim that ILC2 is permissive for dengue virus infection, but the evidence is not strong enough to support this claim. Authors do not demonstrate how DV infects ILC2, neither using freshly isolated ILC2 cells to demonstrate this key issue.

2) Exogenous IFN-beta is able to induce anti-viral ISG genes in ILC2 is nothing unique, as IFN-beta induce ISGs in broad cell types and linages.

3) Authors claim ILC2 has anti-viral activity (figure 4 title). However, no such data are presented in this manuscript. In contrast, authors demonstrate that activated ILC2 enhances dengue virus infection in monocytes (Fig. 5). The anti-viral activity is from IFN-beta, not from ILC2 cells.

Specific comments

1) Figure 1:

a Authors should provide more information of the patients for sample collections. As the immune response is a continuous process, authors should also define which day they collect the samples of each patient.

b Even though healthy control is distinct from DF and DHF, the profiling between DF and DHF is similar. This information demonstrates that ILC2 activation is a general phenomenon in dengue infection.

c Is the gene profiling between HC and DF/DHF is specific in dengue infection? It is possible that ILC2 activation profiling is a general phenomenon in acute viral infection, not dengue specific.

Author should address this question.

d Fig 1d: there is no statistically significant between DF and DHF.

e The presence of dengue RNA in samples may be due to contamination. PI should stain the ILC cells with anti-NS1 and other non-structural protein antibodies and detect the signals by FACS using patients' samples and in vitro culture ILC2 cells.

2) Figure 2

a The infectivity of ILC2 is incredibly high (30%) at MOI = 5 (Fig 2b). If the infectivity is so high, it should be very easy to detect dengue virus-infected ILC2 from patients' sample.

b In addition, PI should use anti-NS1 mAb to confirm the infectivity, not just anti-envelop Ab, because virus may just adsorb on cell surface. To clarify this point, PI should wash ILC2 after incubation, and culture for further ILC2 cells for 48 hours to harvest the supernatant to check virus titer.

c PI should also check the amount of DV RNA levels from the infected ILC2 cells to check whether DV can replicate inside ILC2.

d Heparan sulfate proteoglycans (HSPG) has been reported to enhance DV infection. PI should check whether ILC2 infectivity is via HSPG.

3) Figure 3: it is hard to understand what author want to address.

a In figure 3c, authors show DHF ILC2 express less ISG5 and Stat1 gene. Are these the ILC2 cells freshly isolated from DF and DHF cells?

b Authors should clearly define the y axis of figure 3D. Does PI compare the response of freshly isolated ILC2 cells (or cultured cells) from DF and DHF?

c Fig 3E: The ILC2 cells are from DF or DHF? Authors should compare the results using freshly isolated ILC2 cells from HC, DF and DHF for this experiment.

4) Figure 4:

a) Authors claim PGD4 downregulated antiviral response of ILC2. I do not see any data to demonstrate how ILC2 suppresses DV infection. Figure 4a-4d only shows the effect of PGD2 and IFN-beta to regulate ISG expression in ILC. The conclusion is also confusing.

b) PI use exogenous PGD2 to downregulate DV infection in ILC2. However, authors conclude that endogenously produced PGD2 (by ILC2) plays an important role in dengue infection. How PI can get such kind of conclusion?

c) PGD2 is produced abundantly by mast cells and other cell lineages. What is the amount of PGD2 and LTE4 from DV-infected ILC2 from DF and DHF patients? Compared with other cell types?

5) Figure 5:

a GM-CSF is produced by T cells, macrophages, endothelial cells and fibroblast. Authors should measure the content of GM-CSF in ILC2 culture supernatant, and examine the GM-CSF mRNA in ILC2 to show the levels of GM-CSF from infected ILC2 cells.

UNIVERSITY OF OXFORD

Professor Graham Ogg MA DPhil FRCP
MRC Human Immunology Unit
MRC Weatherall Institute of Molecular
Medicine
Oxford OX3 9DS

E-mail: graham.ogg@ndm.ox.ac.uk
Tel: 44 (0)1865 222334
Fax: 44 (0)1865 222502
www.imm.ox.ac.uk

Dr Shitao Li, PhD

Editorial Board Member

Communications Biology

16th March 2022

**Re: Dengue virus co-opts innate type 2 pathways to escape early control of viral replication
COMMSBIO-21-2798-T**

Thank you very much for your review of the manuscript and for your input and recommendations. On the basis of the advice, we have been able to obtain further samples from dengue-infected individuals and ethnically-matched controls, despite the challenges presented by COVID. We now add new data to address all points raised. The manuscript is undoubtedly improved and so we are very grateful.

We have addressed each point as follows:

Reviewer 1:

Comment 1:

The manuscript “Dengue virus co-opts innate type 2 pathways to escape early control of viral replication” by Fonseka et al., describes that ILC2 are activated during DENV infection, especially in individuals developing SD. They show that ILC2s are activated by PGD2, which also increases the susceptibility of ILC2 directly to DENV infection. PGD2 was found to antagonize the anti-viral effects of type I IFN on ILC2. They also show that ILC2 sups increases monocyte susceptibility to DENV infection via GM-CSF. Authors conclude ILC2 are directly implicated in the immunopathology of dengue, either by increasing monocyte susceptibility to DENV infection or by spreading the virus themselves since are susceptible to infection. They propose that inhibiting PGD2 pathway may be a therapeutic avenue to treat dengue.

The manuscript is technically sound, and all experiments seems to have been performed carefully. A significant novelty of this manuscript is that ILC2 is permissive to DENV infection. It has been demonstrated earlier that lymphocytes (T cells) are also permissive to DENV infection (Silveira et al. J Virol 2018 – not cited). The paper by Silveira et al. is important for their discussion about the role of ILCs to be systemic spreaders of DENV, which seems an immature speculation with the data presented so far. Also, implicating ILC2s conclusively with dengue pathology seems a bit early. At most, an association has been established between ILC2 and SD evolution, which confirms a recently accepted paper that demonstrated that higher frequencies of ILC2 are secreting IL4 and IL13, i.e. they are more activated, in SD cases (Quintino-de Carvalho et al, JID 2021, epub ahead of print – also not cited in the manuscript). Interestingly, Quintino-de-Carvalho found that ILC1, ILC2 and also ILC3 show increased activation levels in SD patients. Type I IFN has also been demonstrated to downmodulate ILC2 (Duerr et al., Nat Immuno 2016 – cited) and that type I IFN counters viral infection is long known.

On the other hand, the manuscript demonstrates that ILC2 might modulate monocytes to be more susceptible to DENV infection and here seems to be the most valuable contribution of the work. How this finding is translated to *in vivo*, still demands further investigation.

Response:

Thank you for these positive comments and careful critique. We have added the recent literature published on dengue and ILC2 (Quintino-de Carvalho et al. 2021, Poonpanichakul T et al, 2021) (Line numbers - 105-108, 362-363). The published data further support the fact that ILC2 are activated in dengue and secrete type 2 cytokines. In addition, we discussed T cell infection data from a recent paper published by Silverira et al. in the discussion section (Line numbers - 395-398). This gave us the opportunity to add pan-T cell, Th2 and Tc2 cell infection data to the manuscript which showed a broader relevance to type 2 responses in dengue infection. Moreover, we have performed comparative infection data between ILC2, Th2 and Tc2 (Line numbers - 393-403) (new figure S5). Specifically, we have presented this in the results section and added additional figures in new figure S4 (on sorting of Th2 and Tc2 cells, line numbers - 549-563) and new figure S5 (T, Th2 Tc2 infection data).

We fully agree that monocyte infection modulation is an important aspect to explore further. However, showing this in the *in vivo* setting may require use of ILC2 knock out murine models to show the effects on infection and outcomes. Development of a compelling murine model of dengue virus infection has been challenging even without the complication of ILC2 depletion (some of the authors have contributed to dengue models and ILC2 depletion models in the literature), because of experimental confounders such as degree of viral replication across different models. Nevertheless, as the reviewer suggests, this would be very important if it can be progressed.

Figure S4.

A)

B)

C)

Figure S5.

Reviewer 2:

Comment 1:

The manuscript entitled “Dengue virus co-opts innate type 2 pathways to escape early control of viral replication” was reviewed. The authors showed ILC2 were activated in dengue infection and found DENV RNA in ILCs. They demonstrated that ILC2 can be infected with DENV in vitro which can be enhanced by CRTH2 activation by PGD2 and blocked by CRTH2 antagonist. Further, they showed supernatants from activated ILC2 facilitate monocytes DENV infection. They suggested that type 2 environment esp. PGD2 decrease ILC2 IFN anti-viral response and support viral dissemination associated with severe disease.

The manuscript was well written and the finding is novel.

Response:

Thank you for these positive comments

Comment 2:

Major comments;

Previous published work on ILC in dengue should be cited and the current results should be discussed with previous work:

- <https://www.frontiersin.org/articles/10.3389/fimmu.2021.599805/full>

- <https://academic.oup.com/jid/advance-article-abstract/doi/10.1093/infdis/jiab312/6298546>

Response:

Thank you. We have discussed these recent publications in the introduction and discussion section.

(Line numbers - 105-108, 362-363, 395-398)

Comment 3:

Although ILC2 was shown to be permissive of DENV infection in this study, their frequency is quite low. The significance /magnitude of its contribution to the overall viremia should be discussed.

Response:

ILC2 are predominantly tissue resident cells and are present in low numbers in the peripheral blood. Although ILC2 circulate at a low percentage in peripheral blood, they act as important mediators of immune responses at the tissue level and promote Th2 cell responses (eg: in helminth infections, as now better discussed in the manuscript). Their presence in skin would be compatible with early exposure to dengue at the point of mosquito-borne infection and so may impact the early control of viral replication. We have accordingly added a paragraph on this in the discussion section. (Line numbers - 375-392)

Comment 4:

Beside monocytes, have the authors investigate their contribution to B cell infection as B cells have recently been shown to be another major source of DENV infection?

Response:

We agree this is an interesting avenue to investigate with the evidence that ILC2 promote early antibody production by B cells in response to viral antigens in the absence of T cells (Drake et al, 2016). Additionally, B cell proliferation and class switching are influenced by type 2 cytokines and so this pathway is likely to be relevant. Interestingly, B cells are relatively rare in human skin and so may not impact the very earliest events, but could certainly contribute to subsequent amplification. We have not investigated this question here, but have added a point to the broader cell types in the discussion. (Line numbers - 450-451)

Comment 5:

Fig 1A-B: Please incorporating donor information in the figure 1A-B. (For example, Fig 1A = using color for donor / shape for severity. Fig 1B - another bar labelling donor)

Response:

We apologise for lack of clarity. We have now added further information to the legend. HC, DF, DHF are 3 different individual groups and are not matched samples from the same donor. Figure 1B specifically highlights the donor information in the lowest portion of the figure. (Line numbers - 132-135, 728-730, modified the legend of Figure 1A).

Comment 6:

Fig 2-4: the authors identified the factors (IL13, PGD2, external IFN β) that affect DENV infectivity on ILC2. How these factors affect the function of ILC2 itself? Does it mean that more infected ILC2 leads to more type 2 cytokine production? How much ILC2 contributes type 2 cytokines to the system overall?

Response:

We have previously shown that IL-33 and PGD2 activate ILC2 and promote type 2 cytokine production (eg Salimi et al 2013, J Exp Med, Hardman et al 2017 Science Transl Med, Hardman et al Science Immunology 2021). We have also previously shown that ILC2 are thought to be early and potent producers of type 2 cytokines themselves, but also amplify type 2 cytokine production from peptide-specific and lipid-specific T cells (Salimi et al 2016 JI, Oliphant et al 2014 Immunity, Hardman et al Science Translational Medicine 2017)

In contrast, IFN- β suppresses type 2 cytokine production and proliferation of ILC2 (Duerr et al – added). We investigated to see whether infected ILC2 produce more type 2 cytokines than mock ILC2, but, there was no difference of cytokine levels between mock and infected ILC2 (new data now added in new figure S2, Line numbers - 203-205). This may indicate that the enhancement of infection is dominated by external activators of ILC2 such as IL-33, PGD2 and LTE4 and less likely to be through dengue infection.

Figure S2.

Comment 7:

Figure 3. It would be interesting to compare DEG with paired convalescence sample from the same patients.

Response:

We agree that it would have been very interesting to study paired acute infection and convalescent samples in individuals. COVID has been devastating in Sri Lanka and so it has been very challenging to get additional samples, but this question will form the basis of future work.

Comment 7:

Fig 5: the authors showed that blocking GM-CSF decreases DENV infection in monocytes. Did ILC2 produce those factors that lead to more infection in monocytes? (they also mention this in discussion on page 14 line 355), I wonder if the authors see different expression of gene coding for GM-CSF (CSF2) from ILC2 RNA-seq data in the context of severity?

Response:

Thank you. Yes. activated ILC2 secrete high amount of GM-CSF with IL-33/IL-25/IL-2 and we add new data to this point (new Figure 5A, B, line numbers - 313-317) and we showed that MMR expression is dependent on GM-CSF secreted by ILC2. We did not observe expression of the *CSF2* gene being significantly altered in the *ex vivo* RNA sequencing analysis of ILC2 in HC, DF and DHF.

Figure 5.

Comment 8:

Minor comments:

Adding tables of DEGs in supplementary material as well as deposited raw data in GEO can be useful for other researchers in the field.

Response:

Thank you, we would welcome editorial advice about whether to include the DEGs in an accompanying spreadsheet or to deposit the data as planned – we are more than happy to do either or both.

Comment 9:

Legend of Fig 2C: include sample size and error bar calculation.

Response:

Thank you. We have added this. (Line numbers - 769-770)

Comment 10:

Graphical abstract showing the roles of PGD2/LTE4/IFN on ILC2 can be helpful

Response:

Thank you. we would welcome editorial advice about whether this would align with figure numbers/space.

Comment 11:

Fig 1: It would be nice if the author can validate that ILC2 were infected *ex-vivo* by FACS or RT-PCR.

Response:

Thank you for this important point. As discussed above, we have performed the *ex vivo* staining of ILC2 from patients with dengue infection. The data confirming ILC2 infection *ex vivo* is presented in new figure 1F, and new figure S1B (Line numbers - 158-160).

Figure 1.

Figure S1.

A)

B)

Reviewer 3:

Comment 1:

This work is to investigate the role of ILC2 in dengue virus infection. However, the data do not support authors claim. The major concerns are as followings:

Authors claim that ILC2 is permissive for dengue virus infection, but the evidence is not strong enough to support this claim. Authors do not demonstrate how DV infects ILC2, neither using freshly isolated ILC2 cells to demonstrate this key issue.

Response:

Thank you for the suggestion. We have now added further evidence supporting the infection of ILC2. As discussed above, we have performed the *ex vivo* staining of ILC2 from patients with dengue infection

(Line numbers - 158-160). The data confirming ILC2 infection *ex vivo* is presented in new figure 1F, and new figure S1B. We also present evidence supporting the role of GM-CSF and mannan (figure 5C-E) and heparin (new figure S3B, line numbers - 210-216) in the infectivity of ILC2.

Figure S3.

Comment 2:

Exogenous IFN-beta is able to induce anti-viral ISG genes in ILC2 is nothing unique, as IFN-beta induce ISGs in broad cell types and linages.

Response:

Thank you, yes, we certainly understand that ISGs are part of a broad cell response. However, this pathway in ILC2 has not been explored in the setting of dengue virus infection. Using human samples during infection has been challenging during the COVID-19 period but has allowed us to make novel observations. We were very struck by the dramatic differences in ISG signals in ILC2 derived from infected individuals and were keen to investigate the underlying mechanisms, of which several are novel. We do hope that the data will be of interest and value to the scientific community, and have implications for dengue pathogenesis and future approaches to treatment.

Comment 3:

Authors claim ILC2 has anti-viral activity (figure 4 title). However, no such data are presented in this manuscript. In contrast, authors demonstrate that activated ILC2 enhances dengue virus infection in monocytes (Fig. 5). The anti-viral activity is from IFN-beta, not from ILC2 cells.

Response:

Thank you. Here we have validated that IFN-beta stimulation induces expression of ISGs in the context of ILC2. This is the likely mechanism where IFN-beta induce control of viral replication. We have stated that it induced an “antiviral state” as opposed to an “antiviral activity/effector function”. We apologise for the lack of clarity and have tried to improve the phrasing (Line numbers – 278, 282).

Comment 3:

Specific comments

Figure 1: Authors should provide more information of the patients for sample collections. As the immune response is a continuous process, authors should also define which day they collect the samples of each patient.

Response:

Thank you. We have added further detail to this in the methods (Line numbers – 465, 516).

Comment 4:

Even though healthy control is distinct from DF and DHF, the profiling between DF and DHF is similar. This information demonstrates that ILC2 activation is a general phenomenon in dengue infection.

Response:

Thank you, yes we entirely agree that there are some shared activation pathways, and as the reviewer points out, we have tried to highlight this in our analysis of the PCA plot (Line numbers – 127-135). However, there are also some dramatic differences such as the ISG pathways observed in the pairwise comparison of healthy control gene expression with DF or DHF. These form the basis of the mechanistic studies presented herein.

Comment 5:

Is the gene profiling between HC and DF/DHF is specific in dengue infection? It is possible that ILC2 activation profiling is a general phenomenon in acute viral infection, not dengue specific. Author should address this question.

Response:

Thank you. We already know that many of the pathways will be shared across other viral infections, but a key question here was the differences between those with DF and DHF in order to try to better understand the determinants of disease severity. The data certainly support differential ISG expression in ILC2, and so we investigated the underlying mechanisms with many novel findings.

Comment 6:

Fig 1d: there is no statistically significant between DF and DHF.

Response:

Thank you, yes that is correct, ILC2 frequency has a higher trend in DHF than DF samples, hence we have not indicated that it is statistically significant. However, as shown, there were many statistically significant differences in the nature of the ILC2 response in those with DF vs DHF (line number 145).

Comment 7:

The presence of dengue RNA in samples may be due to contamination. PI should stain the ILC cells with anti-NS1 and other non-structural protein antibodies and detect the signals by FACS using patients' samples and in vitro culture ILC2 cells.

Response:

Thank you for these suggestions which were very helpful and certainly add to the quality of the manuscript. As above, we have confirmed dengue infection through ex vivo DENV staining in dengue samples (new figure 1F, and new figure S1B). We have also performed NS1 ELISA on supernatants of ILC2 and demonstrated that infected ILC2 secrete DENV NS1 and these data are depicted in new figure 2D (line numbers - 194-199, 666-677).

Figure 2.

Comment 8:

Figure 2 - The infectivity of ILC2 is incredibly high (30%) at MOI = 5 (Fig 2b). If the infectivity is so high, it should be very easy to detect dengue virus-infected ILC2 from patients' sample.

Response:

Thank you. As above, we have confirmed dengue infection through *ex vivo* DENV staining in dengue samples (new figure 1F, and new figure S1B).

Comment 9:

In addition, PI should use anti-NS1 mAb to confirm the infectivity, not just anti-envelop Ab, because virus may just adsorb on cell surface. To clarify this point, PI should wash ILC2 after incubation, and culture for further ILC2 cells for 48 hours to harvest the supernatant to check virus titer.

Response:

Thank you. We have added new data to show that NS1 protein is present in infected ILC2 supernatants (new figure 2D) (line numbers - 194-199). In all experiments after virus inoculation the cells were washed twice with respective media to remove the excess unbound virus and the cells were incubated in media containing IL-2 for 48 hours. We have performed FFA assay on the supernatant to show presence of live virus.

Comment 10:

PI should also check the amount of DV RNA levels from the infected ILC2 cells to check whether DV can replicate inside ILC2.

Response:

Thank you. As above, we have shown DENV infection at RNA level in existing RNA seq *ex vivo* data and at the protein level using FACS staining for DENV E protein (existing *in vitro* data and new data *in vivo* figure 1B and S1B), new data on NS1 in supernatants through ELISA (new figure 2D), and extensively through the FFA assay which showed productive infection.

Comment 11:

Heparan sulphate proteoglycans (HSPG) has been reported to enhance DV infection. PI should check whether ILC2 infectivity is via HSPG.

Response:

Thank you for this suggestion which was very helpful. We have now demonstrated the heparin can indeed reduce dengue viral infection in ILC2 (new Figure S3B). This signified that heparan sulphate can also contribute to DENV attachment in ILC2 (line numbers - 210-216).

Comment 12:

Figure 3: it is hard to understand what author want to address.

Response:

We apologise for our lack of clarity. In figure 3, we have addressed that individuals with DF had prominent type I interferon gene expression which was impaired in those with DHF. Additionally, we

have validated ISG expression in ILC2 through exogenous type I interferon stimulation. We have added further description to the accompanying text (line numbers - 248-253).

Comment 13:

In figure 3c, authors show DHF ILC2 express less ISG5 and Stat1 gene. Are these the ILC2 cells freshly isolated from DF and DHF cells?

Response:

Yes. These cells are sorted from HC, DF and DHF samples. We have added a phrase to clarify that these are from patients' samples (line numbers -251, 265).

Comment 14:

Authors should clearly define the y axis of figure 3D. Does PI compare the response of freshly isolated ILC2 cells (or cultured cells) from DF and DHF?

Response:

Thank you. These are analyses of cultured ILC2, which has been clarified in the axes, and legend and in the accompanying text (figure 3D axes modified, line numbers - 251, 265, 793, 797, 800).

Comment 15:

Fig 3E: The ILC2 cells are from DF or DHF? Authors should compare the results using freshly isolated ILC2 cells from HC, DF and DHF for this experiment.

Response:

These are expanded ILC2 from healthy controls which has been clarified in the axes, and legend and in the accompanying text (figure 3D axes modified, line numbers 251, 265, 793, 797, 800). Although we share the reviewer's excitement about doing these particular analyses *ex vivo*, they are not possible as the cells have to be expanded to sufficient cell numbers first.

Comment 15:

Figure 4: Authors claim PGD2 downregulated antiviral response of ILC2. I do not see any data to demonstrate how ILC2 suppresses DV infection. Figure 4a-4d only shows the effect of PGD2 and IFN-beta to regulate ISG expression in ILC. The conclusion is also confusing.

Response:

Thank you. As above, we apologise for our lack of clarity on this. We have studied an "antiviral state" as opposed to an "antiviral activity/effector function". We have accordingly tried to improve the phrasing (Line numbers - 278, 282).

Comment 15:

PI use exogenous PGD2 to downregulate DV infection in ILC2. However, authors conclude that endogenously produced PGD2 (by ILC2) plays an important role in dengue infection. How PI can get such kind of conclusion?

Response:

We have shown that exogenous PGD2 enhance dengue viral infection (Figure 2 and 4E). We have extended this in the results section to describe relevant literature that ILC2 can produce PGD2 endogenously, and also show here for the first time that ILC2 endogenously produce PGD2 with IL-33 stimulation in the setting of dengue viral infection (Figure 4E, line numbers - 296-298, 304-306). In summary, we show that ILC2 infection is mediated through exogenous and endogenous PGD2.

Comment 16:

PGD2 is produced abundantly by mast cells and other cell lineages. What is the amount of PGD2 and LTE4 from DV-infected ILC2 from DF and DHF patients? Compared with other cell types?

Response:

Thank you. It is clear that mast cells and eosinophils are the major producers of PGD2 and LTE4, and certainly the ILC2 studied herein respond to exogenously provided PGD2 (as described above). Although we confirm previous studies that ILC2 can produce PGD2, and show that this occurs in dengue infection, we do not anticipate that the endogenous source of PGD2 is likely to be a major contributor compared to mast cells and eosinophils. While endogenously produced PGD2 is of interest and useful as validation, it is not a key part of our study (Line numbers – 365-366).

Comment 16:

Figure 5: GM-CSF is produced by T cells, macrophages, endothelial cells and fibroblast. Authors should measure the content of GM-CSF in ILC2 culture supernatant, and examine the GM-CSF mRNA in ILC2 to show the levels of GM-CSF from infected ILC2 cells.

Response:

Thank you. We have now measured the concentration of GM-CSF and IL-13 in activated ILC2 supernatants and added to new figure 5A and 5B (Line numbers – 316-317). As above, we did not observe expression of *CSF2* gene being significantly altered in the *ex vivo* RNA sequencing analysis of ILC2 in HC, DF and DHF.

We are most grateful to the reviewers for sharing valuable insights to undertake further experimental work and provide new data to enhance the quality of the manuscript. Thank you for your further consideration.

Yours sincerely

Graham Ogg

Reviewers' comments:

Reviewer #1 (Remarks to the Author):

I think the authors answered the queries properly.

Reviewer #2 (Remarks to the Author):

Dear authors,

The response to reviewers is satisfactory and the revised manuscript is markedly improve with new important experiments. I have no further comments and think the manuscript is ready to be accepted. Congratulations.

Reviewer #3 (Remarks to the Author):

I am satisfied with most of the responses from the authors except the source of GM-CSF during dengue viral infections. The major source of GM-CSF is from storm cells, T cells and myeloid cells. PI should cautiously discuss the major source of GM-CSF during viral infections. Considering the relatively rare population of ILC2 in vivo, it is hard to believe that the GM-CSF is mainly from ILC-2 cells. PI should have more solid evidence to claim this point.

It has been shown the GM-CSF-derived macrophages are highly susceptible to dengue virus infection, and is the major source of IL_1 and other pro inflammatory cytokines (Wu et al, BLOOD 2013). PI should include this finding when they discuss the critical role of GM-CSF in dengue virus infection.

UNIVERSITY OF OXFORD

Professor Graham Ogg MA DPhil FRCP
MRC Human Immunology Unit
MRC Weatherall Institute of Molecular
Medicine
Oxford OX3 9DS

E-mail: graham.ogg@ndm.ox.ac.uk
Tel: 44 (0)1865 222334
Fax: 44 (0)1865 222502
www.imm.ox.ac.uk

Dr Shitao Li, PhD

Editorial Board Member

Communications Biology

22nd April 2022

Dear Dr Shitao Li

**Re: Dengue virus co-opts innate type 2 pathways to escape early control of viral replication
COMMSBIO-21-2798A**

Thank you to the reviewers for their input and recommendations. We have modified according to the reviewers' suggestions as follows:

Reviewer 1:

I think the authors answered the queries properly.

Response:

Thank you very much.

Reviewer 2:

The response to reviewers is satisfactory and the revised manuscript is markedly improved with new important experiments. I have no further comments and think the manuscript is ready to be accepted. Congratulations.

Response:

Thank you very much.

Reviewer 3:

I am satisfied with most of the responses from the authors except the source of GM-CSF during dengue viral infections. The major source of GM-CSF is from storm cells, T cells and myeloid cells. PI should cautiously discuss the major source of GM-CSF during viral infections. Considering the relatively rare population of ILC2 in vivo, it is hard to believe that the GM-CSF is mainly from ILC-2 cells. PI should have more solid evidence to claim this point.

It has been shown the GM-CSF-derived macrophages are highly susceptible to dengue virus infection, and is the major source of IL-1 and other pro inflammatory cytokines (Wu et al, BLOOD 2013). PI should include this finding when they discuss the critical role of GM-CSF in dengue virus infection.

Response:

Thank you for the suggestions. We have emphasized the wider sources of GM-CSF secretion and added details on the effect of GM-CSF on DENV infection in macrophages. (291-296).

We are most grateful to you and the reviewers for sharing valuable insights to enhance the quality of the manuscript.

Yours sincerely

Graham Ogg